# ETS: Energy-Guided Test-Time Scaling for Training-Free RL Alignment

Xiuyu Li [* 1]   Jinkai Zhang [* 1]   Mingyang Yi [1]   Yu Li [1]   Longqiang Wang [1]   Yue Wang [2]   Ju Fan [1]

## Abstract

Reinforcement Learning (RL) post-training alignment for language models is effective, but also costly and unstable in practice, owing to its complicated training process. To address this, we propose a training-free inference method to sample directly from the optimal RL policy. The transition probability applied to Masked Language Modeling (MLM) consists of a reference policy model and an energy term. Based on this, our algorithm, Energy-Guided Test-Time Scaling (ETS), estimates the key energy term via online Monte Carlo, with a provable convergence rate. Moreover, to ensure practical efficiency, ETS leverages modern acceleration frameworks alongside tailored importance sampling estimators, substantially reducing inference latency while provably preserving sampling quality. Experiments on MLM (including autoregressive models and diffusion language models) across reasoning, coding, and science benchmarks show that our ETS consistently improves generation quality, validating its effectiveness and design. The code is available at https://github.com/sheriyuo/ETS.

## 1. Introduction

Recently, Reinforcement Learning (RL) has become a central paradigm for aligning large language models (LLMs) with human intent (Guo et al., 2025; Ouyang et al., 2022; Li et al., 2026b) in a post-training stage (Kumar et al., 2025). Technically, in the RL stage, the LLM is recognized as a policy model (Sutton et al., 1998), which interacts with a reward model (Schulman et al., 2017; Rafailov et al., 2024; Shao et al., 2024). Despite their empirical success in reasoning (Guo et al., 2025), image generation (Team et al., 2025), natural language tasks (Hou et al., 2024; Luo et al., 2026)

[*]Equal contribution [1]Renmin University of China, Beijing, China [2]Zhongguancun Academy, Beijing, China. Correspondence to: Mingyang Yi <yimingyang@ruc.edu.cn>.

*Proceedings of the 43rd International Conference on Machine Learning*, Seoul, South Korea. PMLR 306, 2026. Copyright 2026 by the author(s).

and agentic workflows (Li et al., 2026a), this paradigm suffers from many fundamental limitations: it requires costly reward modeling (Ouyang et al., 2022; Bai et al., 2022); large-scale human preference data (Rafailov et al., 2024; Zhang et al., 2025b); exhibits unstable training dynamics (Zheng et al., 2025a; Deng et al., 2026) and sensitivity to hyperparameters (Shao et al., 2024); and must be rerun whenever the reward design changes (Huan et al., 2025), leading to substantial computational and human costs.

Theoretically, the existing KL-regularized RL objective of LLM admits a closed-form solution (Shao et al., 2024; Rafailov et al., 2024). However, existing training-based approaches (Zhu et al., 2025) rely on iterative gradient-based optimization to approximate this optimal solution, rather than directly exploiting its structure. This raises a fundamental question: *if the optimal RL distribution is known in closed form, can we sample from it directly at inference time, without any additional training?* By doing so, all the aforementioned issues brought by post-training are resolved.

Inspired by this, we explore inference-time sampling directly from the optimal RL distribution. Theoretically, we show that for the general masked language modeling (MLM) framework (Austin et al., 2021), which includes autoregressive models (ARMs) (Achiam et al., 2023; Yang et al., 2025) and diffusion language models (DLMs) (Nie et al., 2025; Ou et al., 2025), the optimal transition kernel decomposes into two factors: (1) The original transition kernel provided by the reference model. (2) An energy term (Lu et al., 2023; Xu et al., 2025) given by a conditional expectation of exponentiated rewards. This formulation enables *training-free* sampling from the optimal RL policy model.

With the transition formulation, we design Energy-Guided Test-Time Scaling (ETS) to sample from the target optimal distribution without any training process. Moreover, we prove the convergence rate measured by total variation distance (Duchi, 2016) of our method. Notably, our method is conceptually analogous to Monte Carlo Tree Search (MCTS) (Yao et al., 2023) and can be naturally interpreted as a form of Test-Time Scaling (TTS) (Zhang et al., 2025a): additional computation at inference is used to explore multiple candidate continuations and guide the generation toward outcomes with high reward (He et al., 2025). Similar to other TTS methods, the central challenge

of our method is how to perform this exploration efficiently without incurring prohibitive latency. Here, the bottleneck is the energy term in the transition kernel. To resolve this, we design efficient estimators for the energy term and corresponding sampling algorithms tailored to different model families.

To accelerate decoding, for ARMs, we adopt a small proposal model (Yang et al., 2025); for DLMs, we use Fast-dLLM (Wu et al., 2025) with KV caching and parallel decoding. When combined with importance sampling (Tokdar & Kass, 2010), both lightweight proposal models produce Monte Carlo unbiased energy estimators with substantially reduced latency. Theoretically, we prove these estimators yield samples converging to the target distribution.

Empirically, we evaluate ETS across both ARMs and DLMs on reasoning, coding, and science benchmarks (Hendrycks et al., 2021; Rein et al., 2024; Chen, 2021). Our method consistently improves generation quality over standard inference and test-time scaling baselines (Kang et al., 2025; Karan & Du, 2025), surpassing post-trained RL policy without any training. Comprehensive ablations validate our sampling procedures, reward design, and acceleration strategies, demonstrating favorable quality-latency trade-offs.

## 2. Related Work

**Reinforcement Learning (RL).** RL was originally proposed as a framework for trial-and-error learning in sequential decision-making and optimal control (Sutton et al., 1998; Mnih et al., 2015; Watkins & Dayan, 1992; Schulman et al., 2017). Recently, RL has been brought into the LLM post-training, most notably through RLHF and its variants, to align LLM behavior with human preferences and to improve reasoning capabilities (Ouyang et al., 2022; Guo et al., 2025; Zhang et al., 2025b; Su et al., 2025). Common RL-based post-training algorithms for LLMs include PPO, DPO, and GRPO (Schulman et al., 2017; Rafailov et al., 2024; Guo et al., 2025). However, these methods suffer from high retraining costs, reliance on expensive human preference data, strong sensitivity to hyperparameters, and lack of well-defined rewards (Casper et al., 2023; Rafailov et al., 2024).

**Test-Time Scaling (TTS).** TTS improves model performance by allocating additional computation at inference time rather than updating parameters (Snell et al., 2025). Representative TTS techniques include Best-of-N (Nakano et al., 2021), Self-Consistency (Wang et al., 2022; Shafayat et al., 2025), Beam Search (Sutskever et al., 2014), and Monte Carlo Tree Search (MCTS) (Yao et al., 2023; Chang et al., 2025). Beam Search maintains multiple candidate sequences during decoding and approximates maximum a posteriori generation via width-limited pruning, while MCTS enhances reasoning accuracy by sampling diverse solution

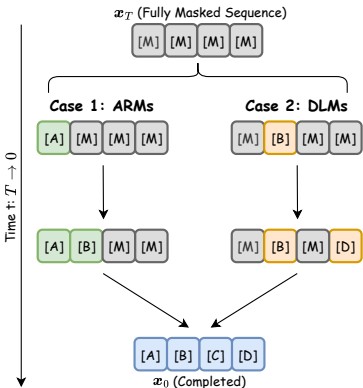

*Figure 1.* Unified MLM framework. Generation is modeled as a backward Markov chain from $x_T$ to $x_0$. Case 1 shows the fixed left-to-right decoding path of ARMs; Case 2 illustrates the flexible, non-sequential unmasking of DLMs.

paths and selecting the most consistent or highest-scoring outcome. Similar to this paper, Power Sampling (Karan & Du, 2025) and Quest (Faria et al., 2024) also sample from RL post-training target distribution by Metropolis–Hastings (MH) algorithm (Chib & Greenberg, 1995), leading to slow generation due to repeated recursive LLM queries. Moreover, (Balashankar et al., 2025; Fei et al., 2025) propose to fit TTS with the target distribution of RL, but their gains heavily rely on training-based RL components rather than offering a general, training-free sampling solution.

Recently, (Dang et al., 2025; Uehara et al., 2024; Singhal et al., 2025) proposed a similar framework to sample from the RLHF post-trained objective by intervening in the generation of trajectories. However, their methods are limited to diffusion models (Ho et al., 2020; Austin et al., 2021; Liu et al., 2025b; Chen et al., 2026). Besides, unlike our method, their methods lack a theoretical convergence rate, and cannot be applied without a verifiable reward function. More importantly, our methods are more efficient than theirs due to our acceleration methods.

**Acceleration.** The goal of inference-time acceleration is to reduce the latency while keeping the model's output distribution essentially unchanged (Zhou et al., 2024; Song et al., 2026). Techniques such as speculative decoding (Leviathan et al., 2023; Li et al., 2025; Hu et al., 2025), quantization (Zheng et al., 2025b), pruning (Ma et al., 2023), and sparse attention (Liu et al., 2025a) are widely used in LLM deployment for this purpose. Since the energy term in our transition kernel can be estimated by a Monte Carlo method, these acceleration methods can be integrated with our method.

## 3. Background

**Masked Language Modeling (MLM) Framework.** We consider an MLM framework that subsumes both ARMs

and DLMs (Austin et al., 2021). Let the vocabulary be denoted by $\mathcal{V}$, and $\boldsymbol{x}_0 = \{x_0^1, \dots, x_0^{d_x}\} \in \mathcal{V}^{d_x}$ be a sequence of length $d_x$ with $\boldsymbol{x}_0^i$ as the $i$-th token. We introduce a special token $[\text{mask}] \in \mathcal{V}$ and define a series of sequences of partially masked states $\{\boldsymbol{x}_t\}_{t=0}^T$ with

$$x_t^i = \begin{cases} x_0^i, & i \notin M_t; \\ [\text{mask}], & i \in M_t. \end{cases} \tag{1}$$

Here $M_t \subseteq \{1, \dots, d_x\}$ is a subset denoting the set of masked positions at step $t$, satisfying $M_s \subset M_t$ for $s < t$ and $M_T = \{1, \cdots, d_x\}$.

In this paper, we only consider the decoding stage of MLM (as shown in Figure 1) . That is, the model progressively generates $\boldsymbol{x}_0$ starting from $\boldsymbol{x}_T$ ($T$ is given in advance). Thus, the mask index decides the decoding order. For ARMs, the mask index $M_t = \{d_x - t + 1, \dots, d_x\}$ corresponds to left-to-right generation. For DLMs, $M_t$ is a random subset of $\{1, \dots, d_x\}$ depending on the logits of predicted tokens.

The decoding process is a backward Markov process with transition kernel

$$p(\boldsymbol{x}_s \mid \boldsymbol{x}_t, \boldsymbol{y}) = \prod_{i=s+1}^t p(\boldsymbol{x}_{i-1} \mid \boldsymbol{x}_i, \boldsymbol{y}) \tag{2}$$

for any $s < t$, where $\boldsymbol{y}$ is a prompt or query. For MLM, the conditional probability $p(\boldsymbol{x}_{i-1} \mid \boldsymbol{x}_i, \boldsymbol{y})$ is modeled by a function call of LLM. For ARMs, the transition $p(\boldsymbol{x}_{i-1} \mid \boldsymbol{x}_i, \boldsymbol{y})$ corresponds to the standard next-token prediction (Achiam et al., 2023). For DLMs, the $p(\boldsymbol{x}_{i-1} \mid \boldsymbol{x}_i, \boldsymbol{y})$ is obtained by modeling the conditional distribution $p(\boldsymbol{x}_0 \mid \boldsymbol{x}_i, \boldsymbol{y})$ to predict $\boldsymbol{x}_0$, and then remask it into $\boldsymbol{x}_{i-1}$. In this paper, for DLMs, we consider the decoding strategy in (Nie et al., 2025) by unmasking the tokens with the top-$K$ predicted probabilities in each decoding step, i.e.,

$$|M_i| - |M_{i-1}| = K, \qquad K = \frac{d_x}{T}. \tag{3}$$

**RL Method.** We consider a general question-answering setting. Given a query $\boldsymbol{y}$, the goal is to generate a response $\boldsymbol{x}_0 = \{x_0^1, \cdots, x_0^{d_x}\} \in \mathcal{V}^{d_x}$. The query–response pair $(\boldsymbol{y}, \boldsymbol{x}_0)$ is evaluated by a reward function $r(\boldsymbol{y}, \boldsymbol{x}_0) \in \mathcal{R}$, where $\mathcal{R}$ may be discrete or continuous.

In this setting, the answering policy is modeled by a LLM $p(\boldsymbol{x}_0 \mid \boldsymbol{y})$. Then, the RL-based post-training methods aim to solve the following KL-regularized optimization problem:

$$\max_{p(\boldsymbol{x}_0 \mid \boldsymbol{y})} \mathbb{E}_{p(\boldsymbol{x}_0 \mid \boldsymbol{y})} \big[ r(\boldsymbol{y}, \boldsymbol{x}_0) - \lambda D_{\text{KL}} \big( p(\boldsymbol{x}_0 \mid \boldsymbol{y}) \,\|\, p_{\text{ref}}(\boldsymbol{x}_0 \mid \boldsymbol{y}) \big) \big], \tag{4}$$

where $p_{\text{ref}}(\boldsymbol{x}_0 \mid \boldsymbol{y})$ denotes a fixed reference model and $\lambda > 0$ is a constant. As can be seen, the goal of RLHF is to maximize the expected reward under the trained policy model, while maintaining its inherent ability by regularizing it to stay close to the reference model.

**Proposition 1.** *(Rafailov et al., 2024) The RLHF objective* (4) *has a closed-form solution*

$$p(\boldsymbol{x}_0 \mid \boldsymbol{y}) = \frac{p_{\text{ref}}(\boldsymbol{x}_0 \mid \boldsymbol{y}) \exp\left(\frac{r(\boldsymbol{y}, \boldsymbol{x}_0)}{\lambda}\right)}{C}, \tag{5}$$

*where $C = \sum_{\boldsymbol{x}_0} p_{\text{ref}}(\boldsymbol{x}_0 \mid \boldsymbol{y}) \exp(r(\boldsymbol{y}, \boldsymbol{x}_0)/\lambda)$ is a normalizing constant.*

## 4. Sample From the Target Distribution

In this section, we present a training-free method to sample from the RL target distribution. Our method has theoretically guaranteed convergence and is applicable to the general MLMs.

### 4.1. Energy Reweighted Backward Transition

Under the unified MLM framework in Section 3, the generation process follows a backward Markov chain $\boldsymbol{x}_T \to \boldsymbol{x}_{T-1} \to \cdots \to \boldsymbol{x}_0$. Therefore, sampling from the optimal distribution (5) reduces to specifying the backward transition kernel $p(\boldsymbol{x}_s \mid \boldsymbol{x}_t, \boldsymbol{y})$ under distribution (5), for each pair of steps $s < t$.

The following proposition derives an explicit expression for this transition by combining the reference model's transition with the expectation of the exponential reward.

**Proposition 2.** *For MLM, for any given query $\boldsymbol{y}$ and response $\boldsymbol{x}_0$, we have*

$$
\begin{aligned}
&p(\boldsymbol{x}_s \mid \boldsymbol{x}_t, \boldsymbol{y}) \\
&= \frac{p(\boldsymbol{x}_t \mid \boldsymbol{x}_s) p(\boldsymbol{x}_s \mid \boldsymbol{x}_0)}{p_{\text{ref}}(\boldsymbol{x}_t \mid \boldsymbol{x}_0)} \cdot \\
&\quad \frac{p_{\text{ref}}(\boldsymbol{x}_s \mid \boldsymbol{y})}{p_{\text{ref}}(\boldsymbol{x}_t \mid \boldsymbol{y})} \frac{\mathbb{E}_{p_{\text{ref}}(\boldsymbol{x}_0 \mid \boldsymbol{y}, \boldsymbol{x}_s)} \left[ \exp\left(\frac{r(\boldsymbol{y}, \boldsymbol{x}_0)}{\lambda}\right) \right]}{\mathbb{E}_{p_{\text{ref}}(\boldsymbol{x}_0 \mid \boldsymbol{y}, \boldsymbol{x}_t)} \left[ \exp\left(\frac{r(\boldsymbol{y}, \boldsymbol{x}_0)}{\lambda}\right) \right]} \\
&\propto p_{\text{ref}}(\boldsymbol{x}_s \mid \boldsymbol{x}_t, \boldsymbol{y}) \mathbb{E}_{p_{\text{ref}}(\boldsymbol{x}_0 \mid \boldsymbol{y}, \boldsymbol{x}_s)} \left[ \exp\left(\frac{r(\boldsymbol{y}, \boldsymbol{x}_0)}{\lambda}\right) \right].
\end{aligned} \tag{6}
$$

Concretely, (6) expresses the target backward transition as a product of two components. (1) The transition kernel $p_{\text{ref}}(\boldsymbol{x}_s \mid \boldsymbol{x}_t, \boldsymbol{y})$ under the reference model; (2) An energy term $\mathbb{E}_{p_{\text{ref}}(\boldsymbol{x}_0 \mid \boldsymbol{y}, \boldsymbol{x}_s)}[\exp(r(\boldsymbol{y}, \boldsymbol{x}_0)/\lambda)]$, which evaluates how likely the state $\boldsymbol{x}_s$ is to lead to high-reward completions under the reference model.

Please notice that the intermediate ratio form in (6) arises from rewriting the RL-optimal joint distribution over the backward trajectory and then isolating the dependence on $\boldsymbol{x}_s$ relative to $\boldsymbol{x}_t$; factors independent of the candidate choice of $\boldsymbol{x}_s$ can be absorbed into the normalization constant.

**Remark 1.** *Notably, (Dang et al., 2025; Uehara et al., 2024; Singhal et al., 2025) propose a similar transition kernel as* (6) *under the continuous-time diffusion model by*

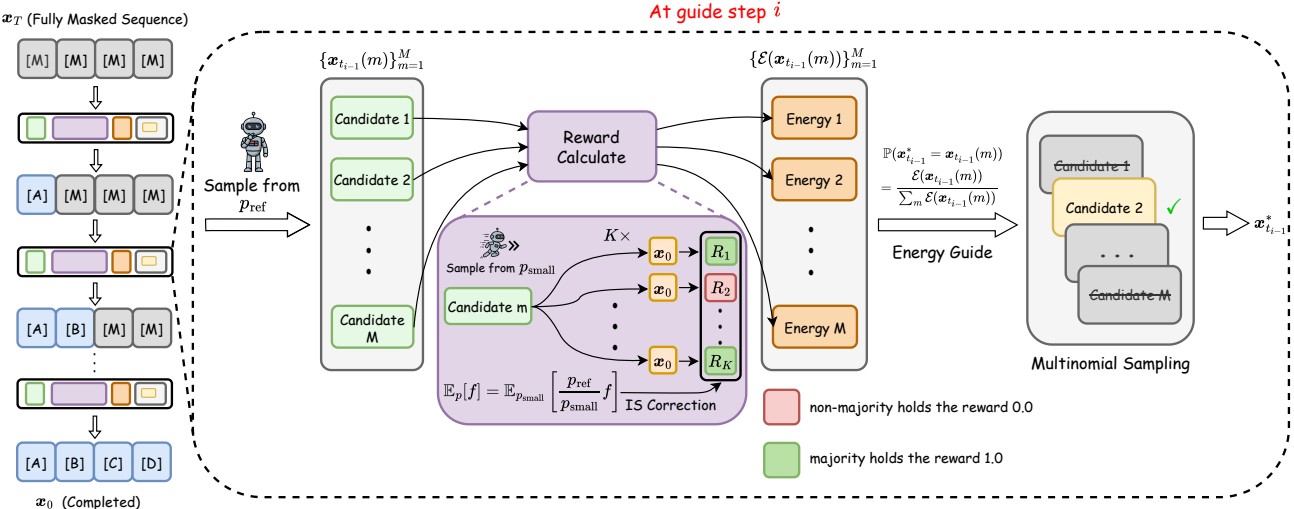

*Figure 2.* Overview of Energy-Guided Test-Time Scaling (ETS). ETS performs iterative guidance on the unified MLM framework. At each guidance step (zoomed-in, right), the algorithm evaluates $M$ candidates. Their associated energy $\mathcal{E}$ is estimated via Monte Carlo method using $K$ independent completions of the corresponding candidate. We utilize an aligned lightweight model $p_{\text{small}}$ with Importance Sampling (IS) correction to accelerate energy estimation while maintaining theoretical consistency with the target optimal distribution.

---

**Algorithm 1** Energy-Guided Test-Time Scaling (ETS): sampling directly from the RL optimal policy.

---

**Input:** Reference model $p_{\text{ref}}$, Guidance steps $I$, Timesteps $\{t_0 = 0, t_1 = \frac{T}{I}, \dots, t_I = T\}$, Initialization $\boldsymbol{x}_T$, Query $\boldsymbol{y}$, Reward function $r$, Candidates number $M$.

**Output:** Sample $\boldsymbol{x}_0$.

1: **for** $i = I, \dots, 1$ **do**
2:     Sampling $\{\boldsymbol{x}_{t_{i-1}}(m)\}_{m=1}^M$ from $p_{\text{ref}}(\boldsymbol{x}_{t_{i-1}} \mid \boldsymbol{y}, \boldsymbol{x}_{t_i})$
3:     Computing $w_m = \frac{\widehat{\mathcal{E}}(\boldsymbol{y}, \boldsymbol{x}_{t_{i-1}}(m))}{\sum_{m=1}^M \widehat{\mathcal{E}}(\boldsymbol{y}, \boldsymbol{x}_{t_{i-1}}(m))}$ under (8).
4:     Selecting $m^* \sim \text{Multinomial}(M, w_1, \cdots, w_m)$
5:     Taking $\boldsymbol{x}_{t_{i-1}} = \boldsymbol{x}_{t_{i-1}}(m^*)$
6: **end for**

---

*continuous-time control theory (Åström, 2012). Compared with their results, our result (6) can be applied to general discrete MLM, and is directly derived from the properties of MLM. (please refer to Appendix A.2 for details)*

### 4.2. Monte Carlo Method

Leveraging the transition kernel (6), we can sample directly from the optimal distribution (5). To implement this, we should estimate the energy term

$$\mathcal{E}(\boldsymbol{y}, \boldsymbol{x}_s) = \mathbb{E}_{p_{\text{ref}}(\boldsymbol{x}_0 \mid \boldsymbol{y}, \boldsymbol{x}_s)} \left[ \exp \left( \frac{r(\boldsymbol{y}, \boldsymbol{x}_0)}{\lambda} \right) \right], \qquad (7)$$

since the transition $p_{\text{ref}}(\boldsymbol{x}_s \mid \boldsymbol{x}_t, \boldsymbol{y})$ can be directly evaluated. Thus, a straightforward way to estimate $\mathcal{E}(\boldsymbol{y}, \boldsymbol{x}_s)$ is

the Monte Carlo estimation (Metropolis & Ulam, 1949)

$$\mathcal{E}(\boldsymbol{y}, \boldsymbol{x}_s) \approx \widehat{\mathcal{E}}(\boldsymbol{y}, \boldsymbol{x}_s) = \frac{1}{K} \sum_{k=1}^K \exp \left( \frac{r(\boldsymbol{y}, \boldsymbol{x}_0(k))}{\lambda} \right),$$
$$\{\boldsymbol{x}_0(k)\}_{k=1}^K \overset{i.i.d.}{\sim} p_{\text{ref}}(\boldsymbol{x}_0 \mid \boldsymbol{y}, \boldsymbol{x}_s). \qquad (8)$$

With this, we propose Algorithm 1 to approximately sample from the optimal distribution via self-normalizing importance sampling (Hammersley, 2013; Cardoso et al., 2022). Intuitively, while the global normalizing constant is intractable, self-normalizing the energies of a sampled batch recovers the relative probabilities among candidates. Sampling from this reweighted batch is equivalent to sampling from the optimal distribution restricted to these candidates.

In Algorithm 1, we guide the generation process with the energy term in steps $t_1, \cdots, t_I$ with $1 \leq I \leq T$. Increasing the number of guidance steps $I$ allows for more frequent re-alignment with the optimal policy, thereby improving the quality of the generated samples. However, this comes at the expense of higher latency, as each guidance step necessitates additional Monte Carlo estimations of the energy term. Notably, by selecting $I = 1$ and $\lambda \to 0$, the Algorithm 1 becomes Best-of-N (Zhang et al., 2025a) since the normalized weights $w_m \propto \mathcal{E}(\boldsymbol{y}, \boldsymbol{x}_{t_{i-1}}(m))$ degenerate to the largest one within the finite candidate set.

Notably, for ETS, ARMs equipped with *batching* and *early-stop* mechanisms decode $M \times K$ sequences in parallel and often terminate early, yielding lower actual per-token cost and fewer generated tokens, as shown in Section 6.2, Appendix B.1 and B.2. Similar to Best-of-N and Beam Search (Wolf et al., 2020), parallel generation algorithms like ETS

better utilize GPU memory and FLOPs, offering greater speed advantages compared to sequential generation methods like Power Sampling (Karan & Du, 2025).

Compared with the existing methods (Dang et al., 2025; Uehara et al., 2024; Singhal et al., 2025), our method demonstrates advantages in applicability, theory, and efficiency. (1) The previous works primarily focus on continuous-time diffusion models, while our unified framework seamlessly covers both DLMs and ARMs. (2) As formally established in Proposition 3, we provide rigorous theoretical guarantees proving that our sampling distribution converges to the optimal target in total variation distance. (3) We address the high latency of existing TTS using the acceleration method with bias correction in Section 5, significantly reducing inference time while maintaining performance. (4) Unlike methods that rely on verifiable reward functions or require fine-tuning value functions, our approach is entirely training-free and operates without such constraints.

### 4.3. Sampling Error Analysis

Algorithm 1 is exact if we can evaluate the energy term in Proposition 2 exactly and draw infinitely many candidates. In practice, however, we approximate the conditional expectation by $\widehat{\mathcal{E}}$ (Section 4.2) and use a finite candidate set of size $M$ to perform the reweighting-and-resampling step. These two approximations induce a deviation between the induced output distribution $q(\boldsymbol{x}_0 \mid \boldsymbol{y})$ of Algorithm 1 and the target distribution $p(\boldsymbol{x}_0 \mid \boldsymbol{y})$.

To quantify this deviation, we measure the discrepancy in total variation distance[1] (Duchi, 2016) and provide an explicit bound that captures how estimation error and finite-$M$ sampling error accumulate across the $I$ guidance steps. The following proposition formalizes this guarantee.

**Proposition 3.** *Suppose that for any given query $\boldsymbol{y}$ and response $\boldsymbol{x}_s$, we have $|\mathcal{E}(\boldsymbol{y}, \boldsymbol{x}_s) - \widehat{\mathcal{E}}(\boldsymbol{y}, \boldsymbol{x}_s)| \leq \epsilon$ for some $\epsilon > 0$. Then for the $\boldsymbol{x}_0 \sim q(\boldsymbol{x}_0 \mid \boldsymbol{y})$ as in Algorithm 1, we have*

$$
\begin{aligned}
&D_{\mathrm{TV}}(q(\boldsymbol{x}_0 \mid \boldsymbol{y}) \parallel p(\boldsymbol{x}_0 \mid \boldsymbol{y})) \\
&\leq I\left(\frac{2\epsilon + h(\epsilon, M, \lambda, D)}{C - \epsilon - h(\epsilon, M, \lambda, D)}\right) + I\epsilon = \widetilde{\mathcal{O}}\left(\frac{I}{\sqrt{M}} + I\epsilon\right),
\end{aligned}
\tag{9}
$$

*where*

$$
h(\epsilon, M, \lambda, D) = \left(\frac{e^{\frac{D}{\lambda}} - e^{-\frac{D}{\lambda}}}{2}\right)\sqrt{\frac{2}{M}\log\left(\frac{2}{\epsilon}\right)}.
\tag{10}
$$

The upper bound reflects the fluctuation induced by finite candidate sampling, scaling as $1/M, \epsilon$ and modulated by the constant $C, D, \lambda$. This indicates that the total variation

---

[1]The total variation distance between distribution $p$ and $q$ is $D_{\mathrm{TV}}(p \parallel q) = \frac{1}{2}\int |p(x) - q(x)|\mathrm{d}x$.

distance grows linearly with $I$, and can be suppressed by increasing $M$ or reducing $\epsilon$. In particular, when $M \to \infty$ and $\epsilon \to 0$, the algorithm converges to the target distribution.

**Remark 2.** *Notably, our bound on the total variation distance is linearly increasing with the guidance steps $I$. This is similar to the results in the continuous diffusion model (Yi et al., 2023; Chen et al., 2023; Yi et al., 2024; Wang et al., 2025), which indicate that the total error accumulates in each denoising step. However, this does not mean that taking $I = 1$ (Best-of-N decoding) presents the best sampling error. This is because we assume the guidance error $\epsilon$ is identical in each guidance step (from $\boldsymbol{x}_{t_i}$ to estimate $\boldsymbol{x}_0$). However, in practice, the guidance error $\epsilon$ may vary across different $\boldsymbol{x}_t$, which may result in a different sampling error bound compared to ours. We leave such a more delicate bound as future work.*

### 4.4. Reward Design

Estimating the energy term $\mathcal{E}(\boldsymbol{y}, \boldsymbol{x}_s)$ in (7) requires a verifiable reward function $r(\boldsymbol{y}, \boldsymbol{x})$, such as correctness against ground truth or human preference scores. However, direct access to the ground-truth reward during inference is impractical (Kang et al., 2025). While prior work trains models to approximate it (Ouyang et al., 2022; Zhang et al., 2025b), this conflicts with our *training-free* objective. We therefore design a training-free *proxy reward* $r(\boldsymbol{y}, \boldsymbol{x})$ that operates without ground-truth, yet induces a reward distribution closely approximating the true one.

We adopt a self-consistency reward derived from the model itself: for each candidate, we sample $K$ completions, extract their final answers, and assign reward 1 to answers matching the majority vote and 0 otherwise. This mechanism favors answers that are statistically consistent across multiple trials, approximating the behavior of a ground-truth reward.

Moreover, we examine various training-free reward candidates, including token-level confidence (logits) (Kadavath et al., 2022), predictive entropy (Kuhn et al., 2023), self-certainty (Kang et al., 2025), and self-consistency (Wang et al., 2022; Shafayat et al., 2025). As shown in Appendix B.4 and D.2, among these uncertainty-based metrics, self-consistency yields a reward distribution closest to the ground-truth reward, providing a tight upper bound on performance attainable with true reward supervision.

## 5. Acceleration Method

In this section, we introduce inference-time acceleration techniques for both ARMs and DLMs. The Monte Carlo estimation of the energy term in (8) incurs substantial computational overhead. To enable practical deployment, we develop efficient estimators that preserve theoretical guarantees while significantly reducing latencies.

**Algorithm 2** ETS-IS (Importance Sampling): accelerate ETS by drawing samples from a lightweight proposal.

**Input:** Reference model $p_{\text{ref}}$, Small model $p_{\text{small}}$, Guidance steps $I$, Timesteps $\{t_0 = 0, t_1 = \frac{T}{I}, \ldots, t_I = T\}$, Initialization $\boldsymbol{x}_T$, Query $\boldsymbol{y}$, Reward function $r$, Candidates number $M$.

**Output:** Sample $\boldsymbol{x}_0$.

1: **for** $i = I, \ldots, 1$ **do**
2:     Sampling $\{\boldsymbol{x}_{t_{i-1}}(m)\}_{m=1}^M$ from $p_{\text{ref}}(\boldsymbol{x}_{t_{i-1}} \mid \boldsymbol{y}, \boldsymbol{x}_{t_i})$
3:     Computing $\widehat{\mathcal{E}}_{\text{small}}(\boldsymbol{y}, \boldsymbol{x}_{t_{i-1}}(m))$ under importance sampling method (11).
4:     Computing $w_m = \frac{\widehat{\mathcal{E}}_{\text{small}}(\boldsymbol{y}, \boldsymbol{x}_{t_{i-1}}(m))}{\sum_{m=1}^M \widehat{\mathcal{E}}_{\text{small}}(\boldsymbol{y}, \boldsymbol{x}_{t_{i-1}}(m))}$
5:     Selecting $m^* \sim \text{Multinomial}(M, w_1, \cdots, w_m)$
6:     Taking $\boldsymbol{x}_{t_{i-1}} = \boldsymbol{x}_{t_{i-1}}(m^*)$
7: **end for**

### 5.1. General Method

Our key observation is that the Monte Carlo estimator only depends on samples drawn from $p_{\text{ref}}$. Therefore, by replacing samples from $p_{\text{ref}}$ with those from a computationally cheaper proposal model, we can significantly reduce the latency while preserving the correctness of the estimator.

By importance sampling (Tokdar & Kass, 2010), we know that for another small proposal model $p_{\text{small}}$ close to $p_{\text{ref}}$, we have

$$
\begin{aligned}
\mathcal{E}(\boldsymbol{y}, \boldsymbol{x}_s) &= \mathbb{E}_{p_{\text{ref}}(\boldsymbol{x}_0 \mid \boldsymbol{y}, \boldsymbol{x}_s)} \left[ \exp\left( \frac{r(\boldsymbol{y}, \boldsymbol{x}_0)}{\lambda} \right) \right] \\
&= \mathbb{E}_{p_{\text{small}}(\boldsymbol{x}_0 \mid \boldsymbol{y}, \boldsymbol{x}_s)} \left[ \frac{p_{\text{ref}}(\boldsymbol{x}_0 \mid \boldsymbol{y}, \boldsymbol{x}_s)}{p_{\text{small}}(\boldsymbol{x}_0 \mid \boldsymbol{y}, \boldsymbol{x}_s)} \exp\left( \frac{r(\boldsymbol{y}, \boldsymbol{x}_0)}{\lambda} \right) \right] \\
&\approx \frac{1}{K} \sum_{k=1}^K \frac{p_{\text{ref}}(\boldsymbol{x}_0(k) \mid \boldsymbol{y}, \boldsymbol{x}_s)}{p_{\text{small}}(\boldsymbol{x}_0(k) \mid \boldsymbol{y}, \boldsymbol{x}_s)} \exp\left( \frac{r(\boldsymbol{y}, \boldsymbol{x}_0(k))}{\lambda} \right) \\
&= \widehat{\mathbb{E}}_{p_{\text{ref}}(\boldsymbol{x}_0 \mid \boldsymbol{y}, \boldsymbol{x}_s)} \left[ \exp\left( \frac{r(\boldsymbol{y}, \boldsymbol{x}_0)}{\lambda} \right) \right] \\
&= \widehat{\mathcal{E}}_{\text{small}}(\boldsymbol{y}, \boldsymbol{x}_s),
\end{aligned}
\tag{11}
$$

where $\{\boldsymbol{x}_0(k)\}_{k=1}^K \overset{i.i.d.}{\sim} p_{\text{small}}(\boldsymbol{x}_0 \mid \boldsymbol{y}, \boldsymbol{x}_s)$. Thus, we can also estimate the energy term by sampling from a cheaper $p_{\text{small}}$. Our acceleration method to approximate the target energy term is summarized in Algorithm 2. Moreover, we have the following proposition to analyze the approximation error in this regime.

**Proposition 4.** *For any given query $\boldsymbol{y}$, noisy response $\boldsymbol{x}_s$ and bounded $r(\boldsymbol{y}, \boldsymbol{x}_0) \in [-D, D]$, if there exists a constant $L > 0$ such that $l(\boldsymbol{x}_0) \leq L$ for all $\boldsymbol{x}_0$, then with probability at least $1 - \delta$ we have*

$$
\left| \mathcal{E}(\boldsymbol{y}, \boldsymbol{x}_s) - \widehat{\mathcal{E}}_{\text{small}}(\boldsymbol{y}, \boldsymbol{x}_s) \right| \leq L e^{\frac{D}{\lambda}} \sqrt{\frac{\log(2/\delta)}{2K}}, \tag{12}
$$

*where*

$$
l(\boldsymbol{x}_0) = \frac{p_{\text{ref}}(\boldsymbol{x}_0 \mid \boldsymbol{y}, \boldsymbol{x}_s)}{p_{\text{small}}(\boldsymbol{x}_0 \mid \boldsymbol{y}, \boldsymbol{x}_s)}. \tag{13}
$$

This proposition shows the trade-off between latency (using $p_{\text{small}}$) and estimation accuracy: a larger $K$ reduces the error but increases inference time, while a poorly matched proposal (large $L$) may require more samples to achieve the same precision. Then, by combining this proposition with Proposition 3, we obtain the following theorem.

**Theorem 1.** *For any given query $\boldsymbol{y}$, noisy response $\boldsymbol{x}_s$, bounded $r(\boldsymbol{y}, \boldsymbol{x}_0) \in [-D, D]$ and $\boldsymbol{x}_0 \sim q(\boldsymbol{x}_0 \mid \boldsymbol{y})$ as in Algorithm 2, if there exists a constant $L > 0$ such that $l(\boldsymbol{x}_0) \leq L$ for all $\boldsymbol{x}_0$, then we have*

$$
\begin{aligned}
&D_{\text{TV}}(q(\boldsymbol{x}_0 \mid \boldsymbol{y}) \parallel p(\boldsymbol{x}_0 \mid \boldsymbol{y})) \\
&\leq I \left( \frac{2\epsilon + g(\epsilon, M, K, L, \lambda, D)}{C - \epsilon - g(\epsilon, M, K, L, \lambda, D)} \right) + 4I \exp\left( -\frac{2K\epsilon^2}{L^2 e^{\frac{2D}{\lambda}}} \right) \\
&= \widetilde{\mathcal{O}} \left( \frac{I}{\sqrt{M}} + \frac{I}{\sqrt{K}} \right),
\end{aligned}
\tag{14}
$$

*where*

$$
\epsilon = \widetilde{\mathcal{O}} \left( \frac{1}{\sqrt{K}} \right), \quad l(\boldsymbol{x}_0) = \frac{p_{\text{ref}}(\boldsymbol{x}_0 \mid \boldsymbol{y}, \boldsymbol{x}_s)}{p_{\text{small}}(\boldsymbol{x}_0 \mid \boldsymbol{y}, \boldsymbol{x}_s)},
$$

$$
g(\epsilon, M, K, L, \lambda, D) = \left( \frac{e^{\frac{D}{\lambda}} - e^{-\frac{D}{\lambda}}}{2} \right) \sqrt{\frac{4K\epsilon^2}{ML^2 e^{\frac{2D}{\lambda}}}} \tag{15}
$$

$$
= \widetilde{\mathcal{O}} \left( \frac{1}{\sqrt{M}} \right).
$$

By choosing $K$ sufficiently large, Algorithm 2 can recover the same asymptotic accuracy as Algorithm 1.

### 5.2. Acceleration for ARMs

For ARMs, we adopt a *scale-aligned* lightweight model as the proposal distribution $p_{\text{small}}$ to accelerate energy estimation. Specifically, we use Qwen3 lightweight models that share the same tokenization with the reference model, which are naturally suitable as $p_{\text{small}}$ due to their similarity. Substituting this choice into Algorithm 2 yields the ETS-IS for ARMs, providing a favorable trade-off between computational efficiency and estimator variance.

We note that alternative approaches like quantization and speculative decoding face practical limitations. Quantized models do not provide operator-level speed advantages at our computational scale, while the speculative decoding method EAGLE-3 (Li et al., 2025) does not support our batching techniques (Appendix B.2) due to tree attention and integration with vLLM (Kwon et al., 2023) inference frameworks. Our ETS-IS approach with distilled models balances implementation feasibility with performance benefits, and we leave the integration with the other acceleration methods as future work (Su et al., 2026; Tang et al., 2026).

### 5.3. Acceleration for DLMs

Unlike ARMs, no small DLM aligned with LLaDA-8B-Instruct is available to serve as $p_{small}$. Therefore, We adopt Fast-dLLM (Wu et al., 2025) as the acceleration scheme for DLMs. Fast-dLLM accelerates inference through two key mechanisms: (1) A block-wise approximate KV cache that reuses similar activations across steps in bidirectional attention. (2) Confidence-aware parallel decoding, which selects high-confidence tokens via a global threshold to increase throughput while preserving quality. We treat the distribution induced by Fast-dLLM as the proposal distribution $p_{small}$ and apply importance sampling to correct the energy estimation, yielding ETS-IS for DLMs.

## 6. Experiments

### 6.1. Experimental Setup

**Evaluation.** We evaluate our methods on a standard suite of reasoning benchmarks covering mathematics (MATH500, GSM8K), coding (HumanEval), and STEM (GPQA). All methods and baselines are evaluated in a single-shot (pass@1) setting, i.e., based on a single final response.

- **MATH500:** The MATH dataset (Hendrycks et al., 2021) consists of competition-level mathematics problems spanning seven categories, containing 12,500 problems with 7,500 training problems and 5,000 test problems. MATH500 is a randomly selected subset of 500 test problems standardized by OpenAI.

- **GSM8K:** GSM8K (Cobbe et al., 2021) is a collection of grade-school-level math word problems designed to evaluate multi-step numerical reasoning. The dataset consists of 8,500 problems in total, with 7,473 training problems and 1,319 test problems.

- **HumanEval:** HumanEval (Chen, 2021) is a benchmark of 164 handwritten programming problems covering algorithms, reasoning, mathematics, and language comprehension. Each problem is associated with an average of 7.7 unit tests, where solving the problem corresponds to passing all unit tests.

- **GPQA:** GPQA (Rein et al., 2024) is a dataset of multiple-choice science questions, designed to require advanced domain reasoning. We evaluate on the GPQA Diamond subset, which consists of 198 questions representing the highest-quality portion of the dataset.

**Models.** To demonstrate the universality of our method across different generative paradigms, we conduct experiments on both ARMs and DLMs. For ARMs, we use Qwen3-1.7B and Qwen3-8B (Yang et al., 2025) in the non-thinking mode as backbone models; For DLMs, we use LLaDA-8B-Instruct (Nie et al., 2025) as the backbone model (LLaDA is only available in 8B).

For all benchmarks, we use lm-eval (Gao et al., 2024) to evaluate both ARMs and DLMs. We refer readers to Appendix C.2 for more detailed hyperparameters.

**Baseline.** The compared baseline methods are categorized into three groups: (1) The standard inference method Base; (2) The TTS methods Beam Search (Wolf et al., 2020), Best-of-N and Power Sampling (Karan & Du, 2025); (3) The RL baseline from Verl (Sheng et al., 2024) and LLaDA-1.5 (Zhu et al., 2025). All of our experiments are reproduced on $4 \times$ H20-3e-141GB GPUs and match the existing SOTA (Karan & Du, 2025; Zhu et al., 2025). We refer readers to Appendix C.1 for more detailed settings.[2]

### 6.2. Main Results

We present our main results in Table 1, which reports accuracy and average latency across tasks. Three key observations emerge:

1. Across different model families, our ETS achieves *substantial* and *universal* boosts in accuracy across different reasoning tasks, consistently matching and even surpassing the performance of the post-trained RL policy (GRPO) without any parameter updates.

2. The acceleration method speeds up the sampling process with slight performance degradation.

3. The latency of our ETS is comparable to the standard TTS methods, e.g., Best-of-N, while achieving significantly better accuracy.

All these results verify the effectiveness and efficiency of ETS methods. Notably, due to the inherently limited parallel decoding speed of LLaDA, its TTS methods incur substantially higher latency than ARMs. Even under this constraint, our acceleration method provides a meaningful reduction in latency while preserving competitive performance.

### 6.3. Ablation Study

We ablate key design choices of our method: total sample count $M \times K$, guidance steps $I$, and the trade-off between latency and accuracy. All other hyperparameters follow the settings in Section 6.2. We present results on Qwen3-8B as a representative ARM; additional ablations on DLM-specific design choices, reward design, temperature sensitivity, and generation length are provided in Appendix D.

---

[2]We do not use ETS-IS for Qwen3-1.7B model since its inference latency is low enough.

*Table 1.* ETS (ours) outperforms other TTS methods and GRPO across model families and tasks. We benchmark the performance of ETS on MATH500, GSM8K, GPQA and HumanEval. Reported accuracies are the best achieved within constrained latency (time and accuracy averaged across datasets), with the best values in each setting bolded.

| Methods | MATH500 | GSM8K | GPQA | HumanEval | Avg | Time |
|---|---|---|---|---|---|---|
| | | | *Qwen3-1.7B* | | | |
| Base | 37.2 | 69.29 | **29.80** | 39.63 | 43.98 | 1× |
| Beam Search | 45.8 | 70.81 | 26.77 | 46.34 | 47.43 | 1.59× |
| Best-of-N | 58.8 | 81.58 | 27.78 | 39.63 | 51.70 | 3.47× |
| Power Sampling | 50.6 | - | 26.30 | 44.51 | - | 40.9× |
| ETS (ours) | **60.4** | **81.88** | **29.80** | **46.95** | **54.76** | 7.18× |
| GRPO | 52.8 | 71.57 | 25.25 | 42.68 | 48.08 | 1× |
| | | | *Qwen3-8B* | | | |
| Base | 60.0 | 89.39 | 34.34 | 58.54 | 60.57 | 1× |
| Beam Search | 61.2 | 89.46 | 32.83 | 69.51 | 63.25 | 1.99× |
| Best-of-N | 65.2 | 94.09 | **38.89** | 67.07 | 66.31 | 5.16× |
| Power Sampling | 70.8 | - | 28.80 | **71.91** | - | 41.8× |
| ETS (ours) | **72.4** | **94.24** | 38.38 | 71.34 | **69.34** | 7.35× |
| ETS-IS (ours) | 66.2 | 91.81 | **38.89** | 68.90 | 66.45 | 5.26× |
| GRPO | 69.2 | 90.98 | 37.88 | 65.85 | 65.98 | 1× |
| | | | *LLaDA-8B-Instruct* | | | |
| Base | 42.2 | 80.36 | 28.28 | 35.98 | 46.71 | 1× |
| Beam Search | 42.8 | 82.71 | 27.78 | 38.41 | 47.93 | 17.6× |
| Best-of-N | 44.2 | 85.82 | 28.28 | 36.59 | 48.90 | 17.4× |
| ETS (ours) | **45.6** | **85.97** | **32.83** | 39.63 | **51.01** | 30.6× |
| ETS-IS (ours) | 43.8 | 85.82 | 28.79 | 39.02 | 49.36 | 16.1× |
| VRPO | 40.2 | 82.03 | 23.74 | **39.63** | 46.40 | 1× |

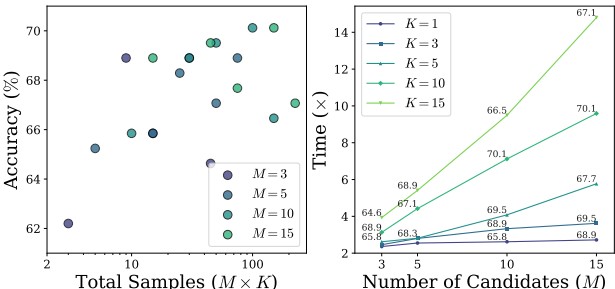

*Figure 3.* Effect of total samples on ETS. We ablate the total samples with Qwen3-8B and plot HumanEval accuracies (left) with corresponding latencies (right) for various sample counts.

**Total Samples.** Figure 3 demonstrates that appropriate $M \times K$ settings yield better results while retaining low latency. In most cases, both accuracy and latency increase as $M \times K$ grows. This is consistent with our theoretical results (Proposition 3). Moreover, we observe that scaling $M$ is more effective for improving accuracy than scaling $K$. According to our empirical observation, selecting $K = 3$ strikes a good balance between accuracy and efficiency.

**Guidance Steps.** In our Algorithm 2, the number of guidance steps is determined by $I$, where a larger $I$ leads to stronger guidance but higher latency. To see its influence, we vary $I$ in Figure 4, which shows that scaling the guidance steps $I$ yields proportional improvements in accuracy alongside increased latency. Owing to this trade-off, we suggest $I = 4$ or $8$ in practice to balance latency and ac-

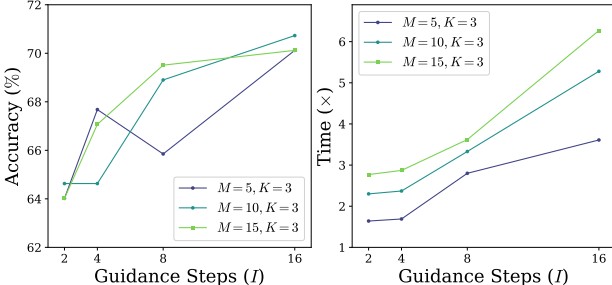

*Figure 4.* Effect of guidance steps on ETS. We evaluate Qwen3-8B on HumanEval (left) with corresponding latencies (right) are reported under various guidance steps.

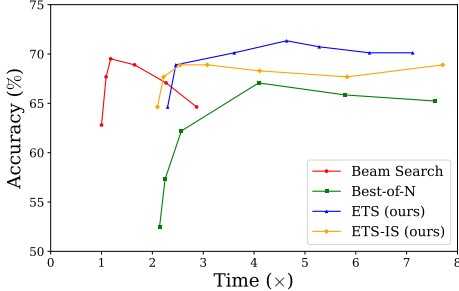

*Figure 5.* Comparisons between TTS methods. We ablate different latencies and plot corresponding Humaneval accuracies with Qwen3-8B, for various training-free TTS methods.

curacy. Moreover, the empirical observations verify that guiding the intermediate decoding steps indeed improves the performance of the reasoning process.

**Time and Accuracy.** We compare ETS with the other TTS methods regarding the trade-off between latency and accuracy. As shown in Figure 5, ETS achieves superior accuracy at comparable latency to other TTS methods. This advantage is pronounced as the computational budget increases. Interestingly, we find that the performance of baseline TTS methods declines significantly with more samples due to verification noise, failing to utilize additional computation effectively (Chow et al., 2024). However, unlike these methods, which naively scale the number of candidates $M$, ETS concurrently scales the prediction range $K$ to produce a smaller set of higher-quality reasoning paths, leading to more robust verification. Moreover, we summarize a practical guideline for our ETS as follows: for constrained budgets, ETS-IS offers efficient initial gains; for larger budgets, standard ETS achieves higher peak performance.

## 7. Conclusion

In this paper, we present ETS, a training-free method for sampling directly from the optimal RL policy in masked language models. By formulating the sampling process as

a Monte Carlo estimation of an energy term, ETS ensures provable convergence to the target distribution and is further accelerated through modern inference frameworks coupled with tailored importance sampling estimators. Experiments across autoregressive and diffusion models on reasoning and coding benchmarks show that ETS exceeds post-trained RL policy, offering an efficient alternative to costly post-training alignment. This work establishes a practical, training-free pathway toward reward-aware generation.

## Acknowledgments

This work was supported by NSF of China Projects (No.62506365).

## Impact Statement

This work demonstrates that ETS enables training-free alignment by directly sampling from the optimal reinforcement learning policy, offering a cost-effective and stable alternative to traditional RL post-training methods. Our framework establishes a foundation for future research on TTS and efficient inference in generative models. This can democratize alignment techniques for resource-constrained settings and reduce the environmental footprint of LLM deployment.

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

# A. Proofs

## A.1. Proof of RL Optimal Distribution

**Proposition 1.** *([Rafailov et al., 2024](#)) The RLHF objective* (4) *has a closed-form solution*

$$p(\boldsymbol{x}_0 \mid \boldsymbol{y}) = \frac{p_{\text{ref}}(\boldsymbol{x}_0 \mid \boldsymbol{y}) \exp\left(\frac{r(\boldsymbol{y}, \boldsymbol{x}_0)}{\lambda}\right)}{C}, \tag{5}$$

*where* $C = \sum_{\boldsymbol{x}_0} p_{\text{ref}}(\boldsymbol{x}_0 \mid \boldsymbol{y}) \exp(r(\boldsymbol{y}, \boldsymbol{x}_0)/\lambda)$ *is a normalizing constant.*

*Proof.* We consider the KL-regularized RL objective

$$\max_{p(\boldsymbol{x}_0 \mid \boldsymbol{y})} \mathbb{E}_{p(\boldsymbol{x}_0 \mid \boldsymbol{y})}\left[r(\boldsymbol{y}, \boldsymbol{x}_0) - \lambda D_{\text{KL}}\left(p(\boldsymbol{x}_0 \mid \boldsymbol{y}) \parallel p_{\text{ref}}(\boldsymbol{x}_0 \mid \boldsymbol{y})\right)\right]. \tag{16}$$

Expanding the expectation and the KL divergence, the objective can be written as

$$\mathcal{J}(p) = \sum_{\boldsymbol{x}_0} p(\boldsymbol{x}_0 \mid \boldsymbol{y}) \, r(\boldsymbol{y}, \boldsymbol{x}_0) - \lambda \sum_{\boldsymbol{x}_0} p(\boldsymbol{x}_0 \mid \boldsymbol{y}) \log \frac{p(\boldsymbol{x}_0 \mid \boldsymbol{y})}{p_{\text{ref}}(\boldsymbol{x}_0 \mid \boldsymbol{y})}. \tag{17}$$

We maximize $\mathcal{J}(p)$ subject to the normalization constraint $\sum_{\boldsymbol{x}_0} p(\boldsymbol{x}_0 \mid \boldsymbol{y}) = 1$. Introducing a Lagrange multiplier $\alpha$, we obtain the Lagrangian

$$\mathcal{L}(p, \alpha) = \sum_{\boldsymbol{x}_0} p(\boldsymbol{x}_0 \mid \boldsymbol{y}) \, r(\boldsymbol{y}, \boldsymbol{x}_0) - \lambda \sum_{\boldsymbol{x}_0} p(\boldsymbol{x}_0 \mid \boldsymbol{y}) \log p(\boldsymbol{x}_0 \mid \boldsymbol{y})$$

$$+ \lambda \sum_{\boldsymbol{x}_0} p(\boldsymbol{x}_0 \mid \boldsymbol{y}) \log p_{\text{ref}}(\boldsymbol{x}_0 \mid \boldsymbol{y}) + \alpha \left(\sum_{\boldsymbol{x}_0} p(\boldsymbol{x}_0 \mid \boldsymbol{y}) - 1\right). \tag{18}$$

Taking the functional derivative with respect to $p(\boldsymbol{x}_0 \mid \boldsymbol{y})$ and setting it to zero yields

$$\frac{\partial \mathcal{L}}{\partial p(\boldsymbol{x}_0 \mid \boldsymbol{y})} = r(\boldsymbol{y}, \boldsymbol{x}_0) - \lambda(\log p(\boldsymbol{x}_0 \mid \boldsymbol{y}) + 1) + \lambda \log p_{\text{ref}}(\boldsymbol{x}_0 \mid \boldsymbol{y}) + \alpha = 0. \tag{19}$$

Solving for $p(\boldsymbol{x}_0 \mid \boldsymbol{y})$ gives

$$p(\boldsymbol{x}_0 \mid \boldsymbol{y}) = \exp\left(\frac{\alpha}{\lambda} - 1\right) p_{\text{ref}}(\boldsymbol{x}_0 \mid \boldsymbol{y}) \exp\left(\frac{r(\boldsymbol{y}, \boldsymbol{x}_0)}{\lambda}\right) = \frac{p_{\text{ref}}(\boldsymbol{x}_0 \mid \boldsymbol{y}) \exp\left(\frac{r(\boldsymbol{y}, \boldsymbol{x}_0)}{\lambda}\right)}{C}, \tag{20}$$

where $C = \frac{1}{\exp\left(\frac{\alpha}{\lambda} - 1\right)}$ is a normalization constant.

Imposing the constraint $\sum_{\boldsymbol{x}_0} p(\boldsymbol{x}_0 \mid \boldsymbol{y}) = 1$, we obtain

$$1 = \frac{\sum_{\boldsymbol{x}_0} p_{\text{ref}}(\boldsymbol{x}_0 \mid \boldsymbol{y}) \exp\left(\frac{r(\boldsymbol{y}, \boldsymbol{x}_0)}{\lambda}\right)}{C}, \tag{21}$$

which implies

$$C = \sum_{\boldsymbol{x}_0} p_{\text{ref}}(\boldsymbol{x}_0 \mid \boldsymbol{y}) \exp\left(\frac{r(\boldsymbol{y}, \boldsymbol{x}_0)}{\lambda}\right). \tag{22}$$

Then we prove our conclusion. $\qquad \square$

## A.2. Proof of Backward Transition

**Proposition 2.** *For both ARMs and DLMs, for any given query* $\boldsymbol{y}$ *and response* $\boldsymbol{x}_0$, *we have*

$$p(\boldsymbol{x}_s \mid \boldsymbol{x}_t, \boldsymbol{y}) = \frac{p(\boldsymbol{x}_t \mid \boldsymbol{x}_s) p(\boldsymbol{x}_s \mid \boldsymbol{x}_0)}{p_{\text{ref}}(\boldsymbol{x}_t \mid \boldsymbol{x}_0)} \frac{p_{\text{ref}}(\boldsymbol{x}_s \mid \boldsymbol{y})}{p_{\text{ref}}(\boldsymbol{x}_t \mid \boldsymbol{y})} \frac{\mathbb{E}_{p_{\text{ref}}(\boldsymbol{x}_0 \mid \boldsymbol{y}, \boldsymbol{x}_s)}\left[\exp\left(\frac{r(\boldsymbol{y}, \boldsymbol{x}_0)}{\lambda}\right)\right]}{\mathbb{E}_{p_{\text{ref}}(\boldsymbol{x}_0 \mid \boldsymbol{y}, \boldsymbol{x}_t)}\left[\exp\left(\frac{r(\boldsymbol{y}, \boldsymbol{x}_0)}{\lambda}\right)\right]} \tag{23}$$

$$\propto p_{\text{ref}}(\boldsymbol{x}_s \mid \boldsymbol{x}_t, \boldsymbol{y}) \mathbb{E}_{p_{\text{ref}}(\boldsymbol{x}_0 \mid \boldsymbol{y}, \boldsymbol{x}_s)}\left[\exp\left(\frac{r(\boldsymbol{y}, \boldsymbol{x}_0)}{\lambda}\right)\right].$$

*Proof.* Let $U_s$ to denote the index of unmasked tokens in $\boldsymbol{x}_s$ with complement $M_s$. By construction of the diffusion process, the clean sequence $\boldsymbol{x}_0$ agrees with $\boldsymbol{x}_s$ on the unmasked positions, i.e., $\boldsymbol{x}_0^{U_s} = \boldsymbol{x}_s^{U_s}$. Consequently, conditioning on $\boldsymbol{x}_s$ is equivalent to conditioning on the fixed values of $\boldsymbol{x}_0^{U_s}$ together with a noise pattern on $M_s$, i.e.,

$$p_{\text{ref}}(\boldsymbol{x}_0 \mid \boldsymbol{y}, \boldsymbol{x}_s) = p_{\text{ref}}(\boldsymbol{x}_0^{M_s} \mid \boldsymbol{y}, \boldsymbol{x}_0^{U_s} = \boldsymbol{x}_s^{U_s}) = p_{\text{ref}}(\boldsymbol{x}_0 \mid \boldsymbol{y}, \boldsymbol{x}_0^{U_s}), \tag{24}$$

Similarly, we note that $p_{\text{ref}}(\boldsymbol{x}_0^{U_s} \mid \boldsymbol{y}) = p_{\text{ref}}(\boldsymbol{x}_s^{U_s} \mid \boldsymbol{y}) = p_{\text{ref}}(\boldsymbol{x}_s \mid \boldsymbol{y})$.

Due to property of total probability and the Markov property of $\boldsymbol{x}_t$,

$$\begin{aligned}
p(\boldsymbol{x}_s \mid \boldsymbol{y}, \boldsymbol{x}_t) &= \sum_{\boldsymbol{x}_0} p(\boldsymbol{x}_s, \boldsymbol{x}_0 \mid \boldsymbol{y}, \boldsymbol{x}_t) \\
&= \sum_{\boldsymbol{x}_0} p(\boldsymbol{x}_s \mid \boldsymbol{x}_0, \boldsymbol{x}_t) p(\boldsymbol{x}_0 \mid \boldsymbol{y}, \boldsymbol{x}_t) \\
&= \sum_{\boldsymbol{x}_0^{M_s}} \sum_{\boldsymbol{x}_0^{U_s}} \frac{p(\boldsymbol{x}_0, \boldsymbol{x}_s, \boldsymbol{x}_t)}{p(\boldsymbol{x}_0, \boldsymbol{x}_t)} \frac{p(\boldsymbol{x}_0, \boldsymbol{x}_t \mid \boldsymbol{y})}{p(\boldsymbol{x}_t \mid \boldsymbol{y})} \\
&= \sum_{\boldsymbol{x}_0^{M_s}} \sum_{\boldsymbol{x}_0^{U_s}} \frac{p(\boldsymbol{x}_t \mid \boldsymbol{x}_s) p(\boldsymbol{x}_s \mid \boldsymbol{x}_0)}{p(\boldsymbol{x}_t \mid \boldsymbol{x}_0)} \frac{p(\boldsymbol{x}_t \mid \boldsymbol{x}_0) p(\boldsymbol{x}_0 \mid \boldsymbol{y})}{p(\boldsymbol{x}_t \mid \boldsymbol{y})} \\
&= \frac{p(\boldsymbol{x}_t \mid \boldsymbol{x}_s) p(\boldsymbol{x}_s \mid \boldsymbol{x}_0)}{p(\boldsymbol{x}_t \mid \boldsymbol{y})} \sum_{\boldsymbol{x}_0^{M_s}} p_{\text{ref}}(\boldsymbol{x}_0^{U_s}, \boldsymbol{x}_0^{M_s} \mid \boldsymbol{y}) \frac{\exp(r(\boldsymbol{y}, \boldsymbol{x}_0))}{C} \\
&= \frac{p(\boldsymbol{x}_t \mid \boldsymbol{x}_s) p(\boldsymbol{x}_s \mid \boldsymbol{x}_0)}{p(\boldsymbol{x}_t \mid \boldsymbol{y})} p_{\text{ref}}(\boldsymbol{x}_0^{U_s} \mid \boldsymbol{y}) \sum_{\boldsymbol{x}_0^{M_s}} p_{\text{ref}}(\boldsymbol{x}_0 \mid \boldsymbol{y}, \boldsymbol{x}_0^{U_s}) \frac{\exp(r(\boldsymbol{y}, \boldsymbol{x}_0))}{C} \\
&= \frac{p(\boldsymbol{x}_t \mid \boldsymbol{x}_s) p(\boldsymbol{x}_s \mid \boldsymbol{x}_0)}{p(\boldsymbol{x}_t \mid \boldsymbol{y})} p_{\text{ref}}(\boldsymbol{x}_0^{U_s} \mid \boldsymbol{y}) \frac{\mathbb{E}_{p_{\text{ref}}(\boldsymbol{x}_0 \mid \boldsymbol{y}, \boldsymbol{x}_0^{U_s})} \left[\exp\left(\frac{r(\boldsymbol{y}, \boldsymbol{x}_0)}{\lambda}\right)\right]}{C} \\
&= \frac{p(\boldsymbol{x}_t \mid \boldsymbol{x}_s) p(\boldsymbol{x}_s \mid \boldsymbol{x}_0)}{p(\boldsymbol{x}_t \mid \boldsymbol{y})} p_{\text{ref}}(\boldsymbol{x}_s \mid \boldsymbol{y}) \frac{\mathbb{E}_{p_{\text{ref}}(\boldsymbol{x}_0 \mid \boldsymbol{y}, \boldsymbol{x}_s)} \left[\exp\left(\frac{r(\boldsymbol{y}, \boldsymbol{x}_0)}{\lambda}\right)\right]}{C}.
\end{aligned} \tag{25}$$

On the other hand,

$$\begin{aligned}
p(\boldsymbol{x}_t \mid \boldsymbol{y}) &= \sum_{\boldsymbol{x}_0^{U_t}} \sum_{\boldsymbol{x}_0^{M_t}} p(\boldsymbol{x}_t, \boldsymbol{x}_0 \mid \boldsymbol{y}) \\
&= p(\boldsymbol{x}_t \mid \boldsymbol{x}_0) \sum_{\boldsymbol{x}_0^{M_t}} \frac{p_{\text{ref}}(\boldsymbol{x}_0^{U_t}, \boldsymbol{x}_0^{M_t} \mid \boldsymbol{y}) \exp\left(\frac{r(\boldsymbol{y}, \boldsymbol{x}_0)}{\lambda}\right)}{C} \\
&= p(\boldsymbol{x}_t \mid \boldsymbol{x}_0) p_{\text{ref}}(\boldsymbol{x}_0^{U_t} \mid \boldsymbol{y}) \sum_{\boldsymbol{x}_0^{M_t}} \frac{p_{\text{ref}}(\boldsymbol{x}_0^{M_t} \mid \boldsymbol{y}, \boldsymbol{x}_0^{U_t}) \exp\left(\frac{r(\boldsymbol{y}, \boldsymbol{x}_0)}{\lambda}\right)}{C} \\
&= p(\boldsymbol{x}_t \mid \boldsymbol{x}_0) p_{\text{ref}}(\boldsymbol{x}_0^{U_t} \mid \boldsymbol{y}) \frac{\mathbb{E}_{p_{\text{ref}}(\boldsymbol{x}_0 \mid \boldsymbol{y}, \boldsymbol{x}_0^{U_t})} \left[\exp\left(\frac{r(\boldsymbol{y}, \boldsymbol{x}_0)}{\lambda}\right)\right]}{C} \\
&= p(\boldsymbol{x}_t \mid \boldsymbol{x}_0) p_{\text{ref}}(\boldsymbol{x}_t \mid \boldsymbol{y}) \frac{\mathbb{E}_{p_{\text{ref}}(\boldsymbol{x}_0 \mid \boldsymbol{y}, \boldsymbol{x}_t)} \left[\exp\left(\frac{r(\boldsymbol{y}, \boldsymbol{x}_0)}{\lambda}\right)\right]}{C}.
\end{aligned} \tag{26}$$

By plugging this into the above equation, we prove our conclusion. $\square$

## A.3. Proof of Total Variance Distance

**Proposition 3.** *Suppose that for any given query $\boldsymbol{y}$ and response $\boldsymbol{x}_t$, we have $|\mathcal{E}(\boldsymbol{y}, \boldsymbol{x}_t) - \widehat{\mathcal{E}}(\boldsymbol{y}, \boldsymbol{x}_t)| \leq \epsilon$ for some $\epsilon > 0$. Then for the $\boldsymbol{x}_0 \sim q(\boldsymbol{x}_0 \mid \boldsymbol{y})$ as in Algorithm 1, we have*

$$D_{\text{TV}}(q(\boldsymbol{x}_0 \mid \boldsymbol{y}) \| p(\boldsymbol{x}_0 \mid \boldsymbol{y})) \leq I\left(\frac{2\epsilon + h(\epsilon, M, \lambda, D)}{C - \epsilon - h(\epsilon, M, \lambda, D)}\right) + I\epsilon = \widetilde{\mathcal{O}}\left(\frac{I}{\sqrt{M}} + I\epsilon\right), \tag{27}$$

*where*

$$h(\epsilon, M, \lambda, D) = \left( \frac{e^{\frac{D}{\lambda}} - e^{-\frac{D}{\lambda}}}{2} \right) \sqrt{\frac{2}{M} \log\left(\frac{2}{\epsilon}\right)}. \tag{28}$$

*Proof.* For any given $i \in \{1, \ldots, I\}$, we construct a distribution $\tilde{q}$ such that $\tilde{q}(\boldsymbol{x}_{t_i} \mid \boldsymbol{y}) = q(\boldsymbol{x}_{t_i} \mid \boldsymbol{y})$ while $\tilde{q}(\boldsymbol{x}_{t_{i-1}} \mid \boldsymbol{y}, \boldsymbol{x}_{t_i}) = p(\boldsymbol{x}_{t_{i-1}} \mid \boldsymbol{y}, \boldsymbol{x}_{t_i})$. Then by the triangle inequality we have

$$D_{\mathrm{TV}}(p(\boldsymbol{x}_{t_{i-1}} \mid \boldsymbol{y}) \parallel q(\boldsymbol{x}_{t_{i-1}} \mid \boldsymbol{y})) \leq D_{\mathrm{TV}}(p(\boldsymbol{x}_{t_{i-1}} \mid \boldsymbol{y}) \parallel \tilde{q}(\boldsymbol{x}_{t_{i-1}} \mid \boldsymbol{y})) + D_{\mathrm{TV}}(\tilde{q}(\boldsymbol{x}_{t_{i-1}} \mid \boldsymbol{y}) \parallel q(\boldsymbol{x}_{t_{i-1}} \mid \boldsymbol{y})). \tag{29}$$

by the triangle's inequality. Due to the definition of total variation distance, we have

$$
\begin{aligned}
D_{\mathrm{TV}}(p(\boldsymbol{x}_{t_{i-1}} \mid \boldsymbol{y}), \tilde{q}(\boldsymbol{x}_{t_{i-1}} \mid \boldsymbol{y})) &= \frac{1}{2} \sum_{\boldsymbol{x}_{t_{i-1}}} |p(\boldsymbol{x}_{t_{i-1}} \mid \boldsymbol{y}) - \tilde{q}(\boldsymbol{x}_{t_{i-1}} \mid \boldsymbol{y})| \\
&= \frac{1}{2} \sum_{\boldsymbol{x}_{t_{i-1}}} \left| \sum_{\boldsymbol{x}_{t_i}} p(\boldsymbol{x}_{t_{i-1}} \mid \boldsymbol{y}, \boldsymbol{x}_{t_i}) p(\boldsymbol{x}_{t_i} \mid \boldsymbol{y}) - \sum_{\boldsymbol{x}_{t_i}} \tilde{q}(\boldsymbol{x}_{t_{i-1}} \mid \boldsymbol{y}, \boldsymbol{x}_{t_i}) \tilde{q}(\boldsymbol{x}_{t_i} \mid \boldsymbol{y}) \right| \\
&= \frac{1}{2} \sum_{\boldsymbol{x}_{t_{i-1}}} \left| \sum_{\boldsymbol{x}_{t_i}} p(\boldsymbol{x}_{t_{i-1}} \mid \boldsymbol{y}, \boldsymbol{x}_{t_i}) [p(\boldsymbol{x}_{t_i} \mid \boldsymbol{y}) - \tilde{q}(\boldsymbol{x}_{t_i} \mid \boldsymbol{y})] \right| \\
&\leq \frac{1}{2} \sum_{\boldsymbol{x}_{t_i}} \sum_{\boldsymbol{x}_{t_{i-1}}} p(\boldsymbol{x}_{t_{i-1}} \mid \boldsymbol{y}, \boldsymbol{x}_{t_i}) |p(\boldsymbol{x}_{t_i} \mid \boldsymbol{y}) - \tilde{q}(\boldsymbol{x}_{t_i} \mid \boldsymbol{y})| \\
&= D_{\mathrm{TV}}(p(\boldsymbol{x}_{t_i} \mid \boldsymbol{y}) \parallel \tilde{q}(\boldsymbol{x}_{t_i} \mid \boldsymbol{y})) \\
&= D_{\mathrm{TV}}(p(\boldsymbol{x}_{t_i} \mid \boldsymbol{y}) \parallel q(\boldsymbol{x}_{t_i} \mid \boldsymbol{y})).
\end{aligned}
\tag{30}
$$

On the other hand, we note that

$$
\begin{aligned}
&D_{\mathrm{TV}}(\tilde{q}(\boldsymbol{x}_{t_{i-1}} \mid \boldsymbol{y}) \parallel q(\boldsymbol{x}_{t_{i-1}} \mid \boldsymbol{y})) \\
&= \sup_{f : \|f\|_\infty \leq 1} \left| \sum_{\boldsymbol{x}_{t_{i-1}}} f(\boldsymbol{x}_{t_{i-1}}) \tilde{q}(\boldsymbol{x}_{t_{i-1}} \mid \boldsymbol{y}) - \sum_{\boldsymbol{x}_{t_{i-1}}} f(\boldsymbol{x}_{t_{i-1}}) q(\boldsymbol{x}_{t_{i-1}} \mid \boldsymbol{y}) \right| \\
&= \sup_{f : \|f\|_\infty \leq 1} \left| \sum_{\boldsymbol{x}_{t_{i-1}}} f(\boldsymbol{x}_{t_{i-1}}) \sum_{\boldsymbol{x}_{t_i}} \tilde{q}(\boldsymbol{x}_{t_{i-1}} \mid \boldsymbol{y}, \boldsymbol{x}_{t_i}) q(\boldsymbol{x}_{t_i} \mid \boldsymbol{y}) - \sum_{\boldsymbol{x}_{t_{i-1}}} f(\boldsymbol{x}_{t_{i-1}}) \sum_{\boldsymbol{x}_{t_i}} q(\boldsymbol{x}_{t_{i-1}} \mid \boldsymbol{y}, \boldsymbol{x}_{t_i}) q(\boldsymbol{x}_{t_i} \mid \boldsymbol{y}) \right| \\
&= \sup_{f : \|f\|_\infty \leq 1} \left| \sum_{\boldsymbol{x}_{t_i}} q(\boldsymbol{x}_{t_i} \mid \boldsymbol{y}) \sum_{\boldsymbol{x}_{t_{i-1}}} f(\boldsymbol{x}_{t_{i-1}}) p(\boldsymbol{x}_{t_{i-1}} \mid \boldsymbol{y}, \boldsymbol{x}_{t_i}) - \sum_{\boldsymbol{x}_{t_i}} q(\boldsymbol{x}_{t_i} \mid \boldsymbol{y}) \sum_{\boldsymbol{x}_{t_{i-1}}} f(\boldsymbol{x}_{t_{i-1}}) q(\boldsymbol{x}_{t_{i-1}} \mid \boldsymbol{y}, \boldsymbol{x}_{t_i}) \right| \\
&= \sup_{f : \|f\|_\infty \leq 1} \left| \sum_{\boldsymbol{x}_{t_i}} q(\boldsymbol{x}_{t_i} \mid \boldsymbol{y}) \left( \mathbb{E}_{p(\boldsymbol{x}_{t_{i-1}} \mid \boldsymbol{y}, \boldsymbol{x}_{t_i})}[f(\boldsymbol{x}_{t_{i-1}})] - \mathbb{E}_{q(\boldsymbol{x}_{t_{i-1}} \mid \boldsymbol{y}, \boldsymbol{x}_{t_i})}[f(\boldsymbol{x}_{t_{i-1}})] \right) \right| \\
&\leq \sup_{f : \|f\|_\infty \leq 1} \sum_{\boldsymbol{x}_{t_i}} q(\boldsymbol{x}_{t_i} \mid \boldsymbol{y}) \left| \mathbb{E}_{p(\boldsymbol{x}_{t_{i-1}} \mid \boldsymbol{y}, \boldsymbol{x}_{t_i})}[f(\boldsymbol{x}_{t_{i-1}})] - \mathbb{E}_{q(\boldsymbol{x}_{t_{i-1}} \mid \boldsymbol{y}, \boldsymbol{x}_{t_i})}[f(\boldsymbol{x}_{t_{i-1}})] \right| \\
&\leq \frac{2\epsilon + h(\epsilon, M, \lambda, D)}{C - \epsilon - h(\epsilon, M, \lambda, D)} + \epsilon,
\end{aligned}
\tag{31}
$$

where the last inequality follows from Lemma 1, and $f$ can be assumed positive without loss of generality. By combining (30) and (31), we get

$$D_{\mathrm{TV}}(p(\boldsymbol{x}_{t_{i-1}} \mid \boldsymbol{y}) \parallel q(\boldsymbol{x}_{t_{i-1}} \mid \boldsymbol{y})) \leq D_{\mathrm{TV}}(p(\boldsymbol{x}_{t_i} \mid \boldsymbol{y}) \parallel q(\boldsymbol{x}_{t_i} \mid \boldsymbol{y})) + \frac{2\epsilon + h(\epsilon, M, \lambda, D)}{C - \epsilon - h(\epsilon, M, \lambda, D)} + \epsilon. \tag{32}$$

Applying this inequality recursively from $i = I$ down to $i = 1$, and noting that $t_I = T$ and $t_0 = 0$, we get

$$D_{\mathrm{TV}}(p(\boldsymbol{x}_0 \mid \boldsymbol{y}) \parallel q(\boldsymbol{x}_0 \mid \boldsymbol{y})) \leq D_{\mathrm{TV}}(p(\boldsymbol{x}_T \mid \boldsymbol{y}) \parallel q(\boldsymbol{x}_T \mid \boldsymbol{y})) + I \left( \frac{2\epsilon + h(\epsilon, M, \lambda, D)}{C - \epsilon - h(\epsilon, M, \lambda, D)} \right) + I\epsilon. \tag{33}$$

Since the initial state $\boldsymbol{x}_T$ is fixed (fully masked), we have $D_{\text{TV}}(p(\boldsymbol{x}_T \mid \boldsymbol{y}), q(\boldsymbol{x}_T \mid \boldsymbol{y})) = 0$. Therefore, we get

$$D_{\text{TV}}(q(\boldsymbol{x}_0 \mid \boldsymbol{y}) \parallel p(\boldsymbol{x}_0 \mid \boldsymbol{y})) \leq I\left(\frac{2\epsilon + h(\epsilon, M, \lambda, D)}{C - \epsilon - h(\epsilon, M, \lambda, D)}\right) + I\epsilon \tag{34}$$

Finally, note that $h(\epsilon, M, \lambda, D) = \widetilde{\mathcal{O}}(1/\sqrt{M})$, so the overall bound is $\widetilde{\mathcal{O}}\left(I/\sqrt{M} + I\epsilon\right)$. $\qquad\square$

**Lemma 1.** *For any given query $\boldsymbol{y}$ and response $\boldsymbol{x}_{t_i}$, if*

$$|\mathcal{E}(\boldsymbol{y}, \boldsymbol{x}_{t_i}) - \widehat{\mathcal{E}}(\boldsymbol{y}, \boldsymbol{x}_{t_i})| \leq \delta \tag{35}$$

*for $\delta = \epsilon$, then*

$$\left|\mathbb{E}_{p(\boldsymbol{x}_{t_{i-1}} \mid \boldsymbol{y}, \boldsymbol{x}_{t_i})}[f(\boldsymbol{x}_{t_{i-1}})] - \mathbb{E}_{q(\boldsymbol{x}_{t_{i-1}} \mid \boldsymbol{y}, \boldsymbol{x}_{t_i})}[f(\boldsymbol{x}_{t_{i-1}})]\right| \leq \frac{2\epsilon + h(\epsilon, M, \lambda, D)}{C - \epsilon - h(\epsilon, M, \lambda, D)} + \epsilon \tag{36}$$

*holds for any positive $f$ with $\|f\|_\infty \leq 1$, where*

$$h(\epsilon, M, \lambda, D) = \left(\frac{e^{\frac{D}{\lambda}} - e^{-\frac{D}{\lambda}}}{2}\right)\sqrt{\frac{2}{M}\log\left(\frac{2}{\epsilon}\right)}. \tag{37}$$

*Proof.* Given $\boldsymbol{y}$ and $\boldsymbol{x}_{t_i}$, we know that

$$\mathbb{E}_{q(\boldsymbol{x}_{t_{i-1}} \mid \boldsymbol{y}, \boldsymbol{x}_{t_i})}[f(\boldsymbol{x}_{t_{i-1}})] = \mathbb{E}_{\boldsymbol{x}_{t_{i-1}}}\left[\sum_{m=1}^{M}\frac{\widehat{\mathcal{E}}(\boldsymbol{y}, \boldsymbol{x}_{t_{i-1}}(m))}{\sum_{m=1}^{M}\widehat{\mathcal{E}}(\boldsymbol{y}, \boldsymbol{x}_{t_{i-1}}(m))}f(\boldsymbol{x}_{t_{i-1}}(m))\right], \tag{38}$$

where $\boldsymbol{x}_{t_{i-1}}(m) \sim p_{\text{ref}}(\boldsymbol{x}_{t_{i-1}} \mid \boldsymbol{y}, \boldsymbol{x}_{t_i})$. Define the event

$$A_{t_i} = \left\{\left|\frac{1}{M}\sum_{m=1}^{M}\mathcal{E}(\boldsymbol{y}, \boldsymbol{x}_{t_{i-1}}(m)) - C\right| \leq h(\epsilon, M, \lambda, D)\right\}. \tag{39}$$

By Lemma 2, we know that $\mathbb{P}\left(A_{t_i}^c\right) \leq \epsilon$. Then, by definition, we know

$$C - \epsilon - h(\epsilon, M, \lambda, D) \leq \frac{1}{M}\sum_{m=1}^{M}\widehat{\mathcal{E}}(\boldsymbol{y}, \boldsymbol{x}_{t_{i-1}}(m)) \leq C + \epsilon + h(\epsilon, M, \lambda, D). \tag{40}$$

holds when event $A_t$ happens. In this regime, by plugging this and (35) into (38), we get

$$\begin{aligned}
&\mathbb{E}_{q(\boldsymbol{x}_{t_{i-1}} \mid \boldsymbol{y}, \boldsymbol{x}_{t_i})}[f(\boldsymbol{x}_{t_{i-1}})] \\
&= \mathbb{E}_{q(\boldsymbol{x}_{t_{i-1}} \mid \boldsymbol{y}, \boldsymbol{x}_{t_i})}[(\mathbf{1}_{A_{t_i}} + \mathbf{1}_{A_{t_i}^c})f(\boldsymbol{x}_{t_{i-1}})] \\
&\leq \mathbb{E}_{\boldsymbol{x}_{t_{i-1}}}\left[\frac{1}{M}\sum_{m=1}^{M}\frac{\mathcal{E}(\boldsymbol{y}, \boldsymbol{x}_{t_{i-1}}(m)) + \epsilon}{C - \epsilon - h(\epsilon, M, \lambda, D)}f(\boldsymbol{x}_{t_{i-1}}(m))\right] + \mathbb{E}_{q(\boldsymbol{x}_{t_{i-1}} \mid \boldsymbol{y}, \boldsymbol{x}_{t_i})}[\mathbf{1}_{A_{t_i}^c}f(\boldsymbol{x}_{t_{i-1}})] \\
&\leq \left(\frac{C}{C - \epsilon - h(\epsilon, M, \lambda, D)}\right)\mathbb{E}_{\boldsymbol{x}_{t_{i-1}}}\left[\frac{1}{M}\sum_{m=1}^{M}\frac{\mathcal{E}(\boldsymbol{y}, \boldsymbol{x}_{t_{i-1}}(m))}{C}f(\boldsymbol{x}_{t_{i-1}}(m))\right] + \frac{\epsilon}{C - \epsilon - h(\epsilon, M, \lambda, D)} + \mathbb{P}(A_{t_i}^c) \\
&\leq \left(\frac{C}{C - \epsilon - h(\epsilon, M, \lambda, D)}\right)\mathbb{E}_{p(\boldsymbol{x}_{t_{i-1}} \mid \boldsymbol{y}, \boldsymbol{x}_{t_i})}[f(\boldsymbol{x}_{t_{i-1}})] + \frac{\epsilon}{C - \epsilon - h(\epsilon, M, \lambda, D)} + \epsilon \\
&\leq \mathbb{E}_{p(\boldsymbol{x}_{t_{i-1}} \mid \boldsymbol{y}, \boldsymbol{x}_{t_i})}[f(\boldsymbol{x}_{t_{i-1}})] + \frac{2\epsilon + h(\epsilon, M, \lambda, D)}{C - \epsilon - h(\epsilon, M, \lambda, D)} + \epsilon.
\end{aligned} \tag{41}$$

Similarly, we can prove the lower bound:

$$\mathbb{E}_{q(\boldsymbol{x}_{t_{i-1}} \mid \boldsymbol{y}, \boldsymbol{x}_{t_i})}[f(\boldsymbol{x}_{t_{i-1}})] \geq \mathbb{E}_{p(\boldsymbol{x}_{t_{i-1}} \mid \boldsymbol{y}, \boldsymbol{x}_{t_i})}[f(\boldsymbol{x}_{t_{i-1}})] - \frac{2\epsilon + h(\epsilon, M, \lambda, D)}{C + \epsilon + h(\epsilon, M, \lambda, D)}. \tag{42}$$

Combining both bounds yields the desired result. $\qquad\square$

**Lemma 2.** *For bounded $r(\boldsymbol{y}, \boldsymbol{x}_0) \in [-D, D]$, we have*

$$\left| \frac{1}{M} \sum_{m=1}^{M} \mathcal{E}(\boldsymbol{y}, \boldsymbol{x}_s(m)) - C \right| \leq \left( \frac{e^{\frac{D}{\lambda}} - e^{-\frac{D}{\lambda}}}{2} \right) \sqrt{\frac{2}{M} \log\left(\frac{2}{\delta}\right)} \tag{43}$$

*holds with probability at least $1 - \delta$.*

*Proof.* Notably, since $r(\boldsymbol{y}, \boldsymbol{x}_0) \in [-D, D]$, we know $\{\mathcal{E}(\boldsymbol{y}, \boldsymbol{x}_s(m))\}_{m=1}^{M}$ are $M$ i.i.d. sub-Gaussian samples with coefficient $(e^{\frac{D}{\lambda}} - e^{-\frac{D}{\lambda}})^2 / 4$. Since $\mathbb{E}_{\boldsymbol{x}_s}[\mathcal{E}(\boldsymbol{y}, \boldsymbol{x}_s)] = C$, by Chernoff's inequality (Duchi, 2016), we prove our conclusion. □

### A.4. Proof of Importance Sampling

**Proposition 4.** *For any given query $\boldsymbol{y}$, noisy response $\boldsymbol{x}_s$ and bounded $r(\boldsymbol{y}, \boldsymbol{x}_0) \in [-D, D]$, if there exists a constant $L > 0$ such that $l(\boldsymbol{x}_0) \leq L$ for all $\boldsymbol{x}_0$, then with probability at least $1 - \delta$ we have*

$$\left| \mathcal{E}(\boldsymbol{y}, \boldsymbol{x}_s) - \widehat{\mathcal{E}}_{\text{small}}(\boldsymbol{y}, \boldsymbol{x}_s) \right| \leq L e^{\frac{D}{\lambda}} \sqrt{\frac{\log(2/\delta)}{2K}}, \tag{12}$$

*where*

$$l(\boldsymbol{x}_0) = \frac{p_{\text{ref}}(\boldsymbol{x}_0 \mid \boldsymbol{y}, \boldsymbol{x}_s)}{p_{\text{small}}(\boldsymbol{x}_0 \mid \boldsymbol{y}, \boldsymbol{x}_s)}. \tag{13}$$

*Proof.* Since $l(\boldsymbol{x}_0) \exp(r(\boldsymbol{y}, \boldsymbol{x}_0)/\lambda) \leq L e^{D/\lambda}$ for all $\boldsymbol{x}_0$, and $\{\boldsymbol{x}_0(k)\}_{k=1}^{K}$ are $K$ i.i.d. samples from $p_{\text{small}}$, we can apply Hoeffding's inequality (Duchi, 2016) for bounded random variables:

$$\mathbb{P}\left( \left| \frac{1}{K} \sum_{k=1}^{K} l(\boldsymbol{x}_0(k)) - \mathbb{E}_{p_{\text{small}}}[l(\boldsymbol{x}_0)] \right| \geq \epsilon \right) \leq 2 \exp\left( -\frac{2K\epsilon^2}{L^2 e^{\frac{2D}{\lambda}}} \right). \tag{44}$$

Setting the right-hand side equal to $\delta$ and solving for $\epsilon$ gives:

$$\epsilon = L e^{\frac{D}{\lambda}} \sqrt{\frac{\log(2/\delta)}{2K}}. \tag{45}$$

Recalling that $\mathbb{E}_{p_{\text{small}}}[l(\boldsymbol{x}_0)] = \mathbb{E}_{p_{\text{ref}}}[\exp(r(\boldsymbol{y}, \boldsymbol{x}_0)/\lambda)] = \mathcal{E}(\boldsymbol{y}, \boldsymbol{x}_s)$, we prove our conclusion. □

**Theorem 1.** *For any given query $\boldsymbol{y}$, noisy response $\boldsymbol{x}_s$, bounded $r(\boldsymbol{y}, \boldsymbol{x}_0) \in [-D, D]$ and $\boldsymbol{x}_0 \sim q(\boldsymbol{x}_0 \mid \boldsymbol{y})$ as in Algorithm 2, if there exists a constant $L > 0$ such that $l(\boldsymbol{x}_0) \leq L$ for all $\boldsymbol{x}_0$, then we have*

$$D_{\text{TV}}(q(\boldsymbol{x}_0 \mid \boldsymbol{y}) \parallel p(\boldsymbol{x}_0 \mid \boldsymbol{y})) \leq I\left( \frac{2\epsilon + g(\epsilon, M, K, L, \lambda, D)}{C - \epsilon - g(\epsilon, M, K, L, \lambda, D)} \right) + 4I \exp\left( -\frac{2K\epsilon^2}{L^2 e^{\frac{2D}{\lambda}}} \right) = \widetilde{\mathcal{O}}\left( \frac{I}{\sqrt{M}} + \frac{I}{\sqrt{K}} \right), \tag{46}$$

*where*

$$\epsilon = \widetilde{\mathcal{O}}\left( \frac{1}{\sqrt{K}} \right), \quad l(\boldsymbol{x}_0) = \frac{p_{\text{ref}}(\boldsymbol{x}_0 \mid \boldsymbol{y}, \boldsymbol{x}_s)}{p_{\text{small}}(\boldsymbol{x}_0 \mid \boldsymbol{y}, \boldsymbol{x}_s)}, \quad g(\epsilon, M, K, L, \lambda, D) = \left( \frac{e^{\frac{D}{\lambda}} - e^{-\frac{D}{\lambda}}}{2} \right) \sqrt{\frac{4K\epsilon^2}{ML^2 e^{\frac{2D}{\lambda}}}} = \widetilde{\mathcal{O}}\left( \frac{1}{\sqrt{M}} \right). \tag{47}$$

*Proof.* For any given query $\boldsymbol{y}$ and noisy response $\boldsymbol{x}_s \in \{\boldsymbol{x}_{t_1}, \ldots \boldsymbol{x}_{t_I}\}$, let us define events

$$A_s = \left\{ \left| \mathcal{E}(\boldsymbol{y}, \boldsymbol{x}_s(m)) - \widehat{\mathcal{E}}_{\text{small}}(\boldsymbol{y}, \boldsymbol{x}_s(m)) \right| \leq \epsilon \right\}, \quad B_s = \left\{ \left| \frac{1}{M} \sum_{m=1}^{M} \mathcal{E}(\boldsymbol{y}, \boldsymbol{x}_s(m)) - C \right| \leq h(\epsilon, M, K, L, \lambda, D) \right\}, \tag{48}$$

where

$$g(\epsilon, M, K, L, \lambda, D) = \left( \frac{e^{\frac{D}{\lambda}} - e^{-\frac{D}{\lambda}}}{2} \right) \sqrt{\frac{4K\epsilon^2}{ML^2 e^{\frac{2D}{\lambda}}}}, \tag{49}$$

By Proposition 4 and Lemma 2, we know that

$$\mathbb{P}(A_s^c) \leq 2 \exp\left( -\frac{2K\epsilon^2}{L^2 e^{\frac{2D}{\lambda}}} \right), \quad \mathbb{P}(B_s^c) \leq 2 \exp\left( -\frac{2K\epsilon^2}{L^2 e^{\frac{2D}{\lambda}}} \right). \tag{50}$$

Let $E_s = A_s \cap B_s$. We have

$$\mathbb{P}(E_s^c) \le \mathbb{P}(A_s^c) + \mathbb{P}(B_s^c) \le 4 \exp\left(-\frac{2K\epsilon^2}{L^2 e^{\frac{2D}{\lambda}}}\right) \tag{51}$$

When $E_s$ occurs, we have

$$C - \epsilon - g(\epsilon, M, K, L, \lambda, D) \le \frac{1}{M}\sum_{m=1}^{M} \widehat{\mathcal{E}}_{\text{small}}(\boldsymbol{y}, \boldsymbol{x}_s(m)) \le C + \epsilon + g(\epsilon, M, K, L, \lambda, D), \tag{52}$$

Following the same argument as in the proof of Proposition 3 and Lemma 1, we have

$$D_{\text{TV}}(q(\boldsymbol{x}_0 \mid \boldsymbol{y}) \| p(\boldsymbol{x}_0 \mid \boldsymbol{y})) \le I\left(\frac{2\epsilon + g(\epsilon, M, K, L, \lambda, D)}{C - \epsilon - g(\epsilon, M, K, L, \lambda, D)}\right) + 4I \exp\left(-\frac{2K\epsilon^2}{L^2 e^{\frac{2D}{\lambda}}}\right). \tag{53}$$

Finally, note that $g(\epsilon, M, K, L, \lambda, D) = \widetilde{\mathcal{O}}(\epsilon\sqrt{K/M})$, so we get $\widetilde{\mathcal{O}}(I\epsilon(1 + \sqrt{K/M}) + I\exp(-K\epsilon^2))$. A common choice is to set $\epsilon = \widetilde{\mathcal{O}}(1/\sqrt{K})$, so the overall bound is $\widetilde{\mathcal{O}}(I/\sqrt{M} + I/\sqrt{K})$. $\qquad\square$

# B. Implementation Details of ETS

## B.1. Computational Complexity of ETS

To quantify our scaling, we analyze the total number of tokens processed by Algorithm 1. Let the guidance block size be $B = d_x/I$. The number of tokens generated over $I$ steps is

$$
\begin{aligned}
N_{\text{tokens}} &= \sum_{i=1}^{I} M(B + K(d_x - iB)) \\
&= Md_x + IMKd_x - \frac{1}{2}(I+1)MKd_x \\
&= M\left(1 + \frac{I-1}{2}K\right)d_x.
\end{aligned}
\tag{54}
$$

Thus, the latency of ETS is approximately $N_{\text{tokens}}/d_x$ times that of a standard single-pass inference, which serves as a worst-case upper bound for both ARMs and DLMs. In practice, ARMs equipped with *batching* and *early-stop* mechanisms decode $\mathcal{O}(MK)$ sequences in parallel and often terminate early, yielding lower actual per-token cost and fewer generated tokens, as shown in Section 6.2.

## B.2. Implementation Details of ETS Pipeline

Figure 6 illustrates our asynchronous evaluation pipeline designed for ARMs with dynamic generation times. For ARMs equipped with an *early-stop* mechanism, the generation time varies across different test examples, which would create idle periods (bubbles) in traditional synchronous evaluation pipelines (lm-eval). Inspired by the asynchronous rollout in AReaL (Fu et al., 2025), we designed this pipeline to fully utilize all of our acceleration techniques. The same pipeline can be applied to DLMs, although the potential for speedup is limited due to their relatively uniform generation times.

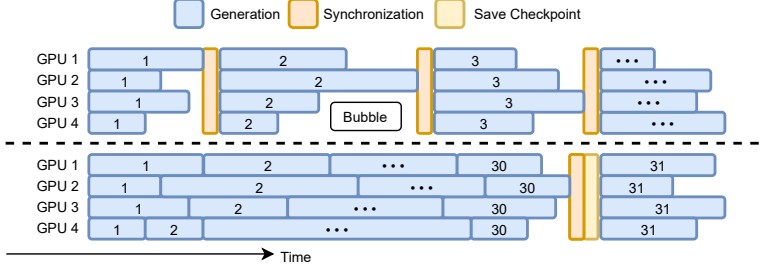

*Figure 6.* Asynchronous ETS evaluation pipeline for ARMs. Up: Traditional synchronous evaluation pipeline. Down: Asynchronous ETS evaluation pipeline. Numbers in the generation process denote the data index being evaluated on each GPU rank.

## B.3. Implementation Details of Reward Calculating

Our reward is implemented via self-consistency: for a set of sampled completions, we first extract each completion's final answer, identify the consensus answer by majority vote, and assign reward 1 to completions matching the consensus and 0 otherwise. To make this procedure more stable and efficient in practice, we adopt an answer caching method.

Concretely, during the entire ETS generation process, whenever we obtain a fully generated sample $x_0$, we extract its final answer $a(x_0)$ and append it to a global cache $\mathcal{A}_{\text{cache}}$.

Later, when estimating energies with self-consistency reward, we draw new completions $\{x_0(i)\}_{i=1}^{MK}$, extract their answers $\{a(x_0(i))\}_{i=1}^{MK}$, and compute the consensus answer over the union of cached and newly obtained answers. Then, we update the global cache $\mathcal{A}_{\text{cache}}$ with these newly obtained answers.

The answer caching method is reasonable because it simply increases the effective sample size used by self-consistency reward, without changing the underlying estimator. Theoretically, as long as the unique optimal answer has the highest probability mass among all possible answers, the probability that self-consistency selects an incorrect answer decays exponentially with the sample size.

Let $A$ denote the extracted final answer from a single completion, taking values in a large discrete set $\mathcal{A}$. Let $a^\star \in \mathcal{A}$ be the unique correct answer with probability

$$\mathbb{P}(A = a^\star) = p, \qquad \mathbb{P}(A = a) = q_a \ (a \neq a^\star), \qquad \sum_{a \neq a^\star} q_a = 1 - p. \tag{55}$$

Assume that $p > q_{\max}$ where $q_{\max} = \max_{a \neq a^\star} q_a$. Given $N$ i.i.d. samples $\{A_i\}_{i=1}^N$, self-consistency predicts the consensus answer

$$\hat{a} = \arg\max_{a \in \mathcal{A}} C_a, \qquad C_a = \sum_{i=1}^N \mathbb{I}[A_i = a]. \tag{56}$$

**Theorem 2** (Exponential decay of self-consistency voting error). *Under the assumption $p > q_{\max}$ and i.i.d. sampling, the probability that self-consistency selects an incorrect answer satisfies*

$$\mathbb{P}(\hat{a} \neq a^\star) \leq \sum_{a \neq a^\star} \exp\left(-\frac{N(p - q_a)^2}{2}\right) \leq (|\mathcal{A}| - 1) \exp\left(-\frac{N(p - q_{\max})^2}{2}\right), \tag{57}$$

*where $|\mathcal{A}|$ can be replaced by the number of answers with nonzero probability mass if $\mathcal{A}$ is large.*

*Proof.* Self-consistency makes an error if and only if there exists some $a \neq a^\star$ whose vote count is no smaller than that of $a^\star$, i.e.,

$$\{\hat{a} \neq a^\star\} \subseteq \bigcup_{a \neq a^\star} \{C_a \geq C_{a^\star}\}. \tag{58}$$

By the union bound,

$$\mathbb{P}(\hat{a} \neq a^\star) \leq \sum_{a \neq a^\star} \mathbb{P}(C_a \geq C_{a^\star}). \tag{59}$$

Fix any $a \neq a^\star$ and define for each sample

$$Y_i^{(a)} = \mathbb{I}[A_i = a^\star] - \mathbb{I}[A_i = a], \qquad Y_i^{(a)} \in \{-1, 0, 1\} \subset [-1, 1]. \tag{60}$$

Then

$$\sum_{i=1}^N Y_i^{(a)} = C_{a^\star} - C_a, \qquad \mathbb{E}[Y_i^{(a)}] = p - q_a > 0. \tag{61}$$

Therefore,

$$\mathbb{P}(C_a \geq C_{a^\star}) = \mathbb{P}\left(\sum_{i=1}^N Y_i^{(a)} \leq 0\right) = \mathbb{P}\left(\frac{1}{N}\sum_{i=1}^N Y_i^{(a)} - (p - q_a) \leq -(p - q_a)\right). \tag{62}$$

Applying Hoeffding's inequality for independent bounded random variables in $[-1, 1]$ yields

$$\mathbb{P}(C_a \geq C_{a^\star}) \leq \exp\left(-\frac{N(p - q_a)^2}{2}\right). \tag{63}$$

Summing over all $a \neq a^\star$ proves the first inequality. The second inequality follows from $p - q_a \geq p - q_{\max}$ for all $a \neq a^\star$. $\square$

### B.4. The Candidates of Reward Designs

**Logits.** The log-probability reward measures how likely a sampled response trajectory is under the reference policy. Intuitively, responses with higher average token log-probability lie in higher-density regions of $p_{\mathrm{ref}}(x_0 \mid y)$, and thus reflect stronger model confidence in its own generation. We define the reward as the average per-token log-probability along the trajectory:

$$r_{\mathrm{logp}}(\boldsymbol{y}, \boldsymbol{x}_0) = \frac{1}{d_x} \sum_{i=1}^{d_x} \log p_{\mathrm{ref}}\left(x_0^i \mid \boldsymbol{y}, \boldsymbol{x}_0^{<i}\right). \tag{64}$$

**Entropy.** The entropy reward captures predictive uncertainty at each generation step by inspecting the next-token distribution $p_{\mathrm{ref}}(\cdot \mid \boldsymbol{y}, \boldsymbol{x}_0^{<i})$. A more concentrated distribution indicates higher certainty, while a flatter distribution indicates higher uncertainty. To align the direction of optimization with confidence, we use the *negative entropy* and average it over time:

$$r_{\mathrm{ent}}(\boldsymbol{y}, \boldsymbol{x}_0) = \frac{1}{d_x} \sum_{i=1}^{d_x} \sum_{v \in V} p_{\mathrm{ref}}\left(v \mid \boldsymbol{y}, \boldsymbol{x}_0^{<i}\right) \log p_{\mathrm{ref}}\left(v \mid \boldsymbol{y}, \boldsymbol{x}_0^{<i}\right). \tag{65}$$

**Self-Certainty.** Self-certainty quantifies how far the model's next-token distribution deviates from the uniform distribution. When the predictive distribution is strongly peaked, it departs more from uniformity, indicating higher confidence. Following the standard definition, we compute an average token-level score based on the log-likelihood ratio against a uniform baseline:

$$r_{\mathrm{sc}}(\boldsymbol{y}, \boldsymbol{x}_0) = -\frac{1}{d_x |V|} \sum_{i=1}^{d_x} \sum_{v \in V} \log\left(|V| \cdot p_{\mathrm{ref}}(v \mid \boldsymbol{y}, \boldsymbol{x}_0^{<i})\right). \tag{66}$$

Up to an additive constant, this is equivalent to the divergence $D_{\mathrm{KL}}(U \parallel p_{\mathrm{ref}}(\cdot \mid \boldsymbol{y}, \boldsymbol{x}_0^{<i}))$, where $U$ denotes the uniform distribution over $V$.

**Self-Consistency.** Self-consistency uses agreement among multiple independent samples as a proxy for reliability. Given the same conditioning input, we sample $K$ i.i.d. responses $\{\boldsymbol{x}_0(k)\}_{k=1}^K$ from $p_{\mathrm{ref}}(\boldsymbol{x}_0 \mid \boldsymbol{y})$ (or from $p_{\mathrm{ref}}(\boldsymbol{x}_0 \mid \boldsymbol{y}, \boldsymbol{x}_s)$ when conditioning on an intermediate state $\boldsymbol{x}_s$), and extract a discrete final answer $a(k)$ from each completion (e.g., via a task-specific parsing function). We then define the consensus answer by majority voting and reward samples that match the consensus:

$$\begin{aligned}
\boldsymbol{x}_0(k) &\overset{\mathrm{i.i.d.}}{\sim} p_{\mathrm{ref}}(\boldsymbol{x}_0 \mid \boldsymbol{y}), \quad k = 1, \ldots, K, \\
a(k) &= \mathrm{Extract}(\boldsymbol{x}_0(k)), \qquad a^\star = \mathrm{mode}\left(\{a(k)\}_{k=1}^K\right), \\
r_{\mathrm{cons}}(\boldsymbol{y}, \boldsymbol{x}_0(k)) &= \mathbb{I}[a(k) = a^\star].
\end{aligned} \tag{67}$$

## C. Implementation Details of Experiments

### C.1. Baseline Settings

For the Base method, we follow the original settings from (Yang et al., 2025) and (Nie et al., 2025).

Best-of-N is naturally integrated into our ETS framework as a special case, with detailed hyperparameters provided in Appendix C.2.

For Beam Search, we use the standard implementation in the transformers (Wolf et al., 2020) to evaluate ARMs with original temperature $t = 0.7$ (refer to Appendix D.3), leveraging its parallel decoding via batching. For DLMs, we implement beam search ourselves; however, due to their iterative generation nature, DLMs cannot be accelerated via batching in the same way as ARMs.

For Power Sampling (Karan & Du, 2025), we retain the original $\alpha = 0.25$, $N_{\mathrm{MCMC}} = 10$ and $B = d_x/16$ settings, but reduce the generation length $d_x$ from 3072 to 512 due to its *substantial* latency of $(N_{\mathrm{MCMC}} d_x)/(4B) = 40\times$, which places it in a different computational regime compared to other TTS methods. Furthermore, unlike general evalutaion frameworks such as lm-eval, Power Sampling requires a hand-crafted, task-specific verifier, which precludes its evaluation on GSM8K.

For the GRPO baseline, we post-trained Qwen3-1.7B and Qwen3-8B in the non-thinking mode on the MATH dataset for 205 training steps, with greedy decoding for evalution. For DLMs, we directly use the results from LLaDA-1.5 fine-tuned with VRPO.

## C.2. Hyperparameter Settings of ETS

Table 2, 3 and 4 list the hyperparameters used in the experiments summarized in Table 1. For ARMs, we configure the generation length $d_x = 512$ (ablation study in Appendix D.4). For DLMs, we retain the original dynamic generation lengths, which vary across datasets. The selection of hyperparameters follows the findings from the ablation study in Section 6.3 and Appendix D. For the KL regularization coefficient $\lambda$, we fix it to 0.1.

Notably, ETS-IS achieves comparable or even superior accuracy-latency trade-offs to the original ETS under certain hyperparameter settings. Our hyperparameter selection thus proceeds in two stages: we first identify a strong ETS configuration, then adjust it with reference to the latency achieved by Best-of-N while preserving accuracy. For ARMs, we employ Qwen3-1.7B as $p_{\text{small}}$ to accelerate energy estimation for Qwen3-8B. We do not evaluate ETS-IS on Qwen3-1.7B, because its base inference speed is already sufficiently fast and the extra overhead of computing logits for importance sampling would outweigh any acceleration benefit. For DLMs, the introduction of Fast-dLLM provides substantial acceleration, so we directly adopt the ETS hyperparameters.

*Table 2.* Hyperparameter settings in Table 1 for Qwen3-1.7B model. $t$ is the *softmax temperature* for LLM generation. $N_{\text{tokens}}/d_x$ provides an upper bound on the *multiplicative latency* relative to a standard single-pass (base) inference, which has $N_{\text{tokens}}/d_x = 1$.

| Methods | Benchmarks | $M$ | $K$ | $I$ | $B$ | $t$ | $N_{\text{tokens}}/d_x$ | Time |
|---|---|---|---|---|---|---|---|---|
| Beam Search | MATH500 | 20 | - | - | - | 0.7 | 20 | 1.47× |
| | GSM8K | 20 | - | - | - | 0.7 | 20 | 1.63× |
| | GPQA | 20 | - | - | - | 0.7 | 20 | 1.89× |
| | HumanEval | 20 | - | - | - | 0.7 | 20 | 1.24× |
| Best-of-N | MATH500 | 50 | 1 | 1 | 512 | 0.7 | 50 | 4.73× |
| | GSM8K | 20 | 1 | 1 | 512 | 1.5 | 20 | 2.97× |
| | GPQA | 30 | 1 | 1 | 512 | 0.5 | 30 | 3.87× |
| | HumanEval | 50 | 1 | 1 | 512 | 0.25 | 50 | 3.30× |
| ETS | MATH500 | 15 | 3 | 8 | 64 | 0.7 | 172.5 | 8.94× |
| | GSM8K | 15 | 3 | 8 | 64 | 1.5 | 172.5 | 6.92× |
| | GPQA | 5 | 3 | 8 | 64 | 1.5 | 57.5 | 5.94× |
| | HumanEval | 10 | 10 | 8 | 64 | 0.25 | 360 | 5.53× |

*Table 3.* Hyperparameter settings in Table 1 for Qwen3-8B model. $t$ is the *softmax temperature* for LLM generation. $N_{\text{tokens}}/d_x$ provides an upper bound on the *multiplicative latency* relative to a standard single-pass (base) inference, which has $N_{\text{tokens}}/d_x = 1$.

| Methods | Benchmarks | $M$ | $K$ | $I$ | $B$ | $t$ | $N_{\text{tokens}}/d_x$ | Time |
|---|---|---|---|---|---|---|---|---|
| Beam Search | MATH500 | 20 | - | - | - | 0.7 | 20 | 1.57× |
| | GSM8K | 20 | - | - | - | 0.7 | 20 | 1.88× |
| | GPQA | 50 | - | - | - | 0.7 | 50 | 4.48× |
| | HumanEval | 20 | - | - | - | 0.7 | 20 | 1.18× |
| Best-of-N | MATH500 | 50 | 1 | 1 | 512 | 0.7 | 50 | 6.09× |
| | GSM8K | 30 | 1 | 1 | 512 | 1.5 | 30 | 4.89× |
| | GPQA | 30 | 1 | 1 | 512 | 1.5 | 30 | 5.48× |
| | HumanEval | 50 | 1 | 1 | 512 | 0.25 | 50 | 4.09× |
| ETS | MATH500 | 10 | 3 | 8 | 64 | 0.7 | 115 | 9.85× |
| | GSM8K | 15 | 3 | 4 | 128 | 1.5 | 82.5 | 6.21× |
| | GPQA | 15 | 3 | 8 | 64 | 0.25 | 172.5 | 5.71× |
| | HumanEval | 20 | 3 | 8 | 64 | 0.25 | 230 | 4.67× |
| ETS-IS | MATH500 | 15 | 3 | 4 | 128 | 0.7 | 82.5 | 5.88× |
| | GSM8K | 15 | 3 | 4 | 128 | 1.5 | 82.5 | 5.33× |
| | GPQA | 15 | 3 | 8 | 64 | 0.25 | 172.5 | 5.50× |
| | HumanEval | 10 | 3 | 8 | 64 | 0.25 | 115 | 2.55× |

*Table 4.* Hyperparameter settings in Table 1 for LLaDA-8B-Instruct model. $d_x$ is the answer length that LLaDA is allowed to generate for the final output sequence. Block length denotes the number of tokens generated per block under LLaDA's semi-autoregressive strategy. $t$ is the *softmax temperature* for LLM generation. $N_{\text{tokens}}/d_x$ provides an upper bound on the *multiplicative latency* relative to a standard single-pass (base) inference, which has $N_{\text{tokens}}/d_x = 1$.

| Methods | Benchmarks | $d_x$ | Block length | $M$ | $K$ | $I$ | $B$ | $t$ | $N_{\text{tokens}}/d_x$ | Time |
|---|---|---|---|---|---|---|---|---|---|---|
| Beam Search | MATH500 | 256 | 32 | 20 | - | - | - | 0.5 | 20 | 9.38× |
| | GSM8K | 256 | 8 | 20 | - | - | - | 1.0 | 20 | 22.5× |
| | GPQA | 64 | 8 | 20 | - | - | - | 0.5 | 20 | 13.9× |
| | HumanEval | 512 | 32 | 20 | - | - | - | 0.25 | 20 | 10.5× |
| Best-of-N | MATH500 | 256 | 32 | 20 | 1 | 1 | 256 | 0.5 | 20 | 14.9× |
| | GSM8K | 256 | 8 | 15 | 1 | 1 | 256 | 1.0 | 15 | 14.7× |
| | GPQA | 64 | 8 | 20 | 1 | 1 | 128 | 0.5 | 20 | 22.9× |
| | HumanEval | 512 | 32 | 20 | 1 | 1 | 512 | 0.25 | 20 | 24.5× |
| ETS | MATH500 | 256 | 32 | 15 | 1 | 4 | 64 | 0.5 | 37.5 | 20.6× |
| | GSM8K | 256 | 8 | 10 | 1 | 4 | 64 | 1.0 | 25 | 30.9× |
| | GPQA | 64 | 8 | 5 | 3 | 8 | 16 | 0.5 | 57.5 | 57.1× |
| | HumanEval | 512 | 32 | 5 | 3 | 8 | 64 | 0.25 | 57.5 | 47.7× |
| ETS-IS | MATH500 | 256 | 32 | 5 | 3 | 8 | 32 | 0.5 | 57.5 | 7.24× |
| | GSM8K | 256 | 8 | 10 | 1 | 4 | 64 | 1.0 | 25 | 17.2× |
| | GPQA | 64 | 8 | 5 | 3 | 8 | 16 | 0.5 | 57.5 | 19.1× |
| | HumanEval | 512 | 32 | 5 | 3 | 8 | 64 | 0.25 | 57.5 | 21.6× |

# D. Additional Experimental Results

## D.1. More Results on DLMs

Following the results in Section 6.3, we conduct similar ablation studies on DLMs, including the variations over $M \times K$, guidance step $I$, and accuracies under various latencies. For time considerations, we use 200 randomly selected examples from the GSM8K dataset for this ablation on LLaDA.

As shown in Figure 7 and Figure 8, we can obtain the same conclusions as in Section 6.3 on ARMs. Specifically, an appropriate choice of $M \times K$ can improve performance while maintaining low latency. Both accuracy and latency increase as $M \times K$ increases, and scaling $M$ is empirically more effective than scaling $K$. For the guidance schedule, increasing the number of guide steps $I$ strengthens guidance but incurs a higher latency. Therefore, we recommend selecting $I$ to balance accuracy and computational efficiency. For latency and accuracy, ETS achieves higher accuracy at comparable latency than other TTS baselines, and its advantage becomes more pronounced as the computational budget increases.

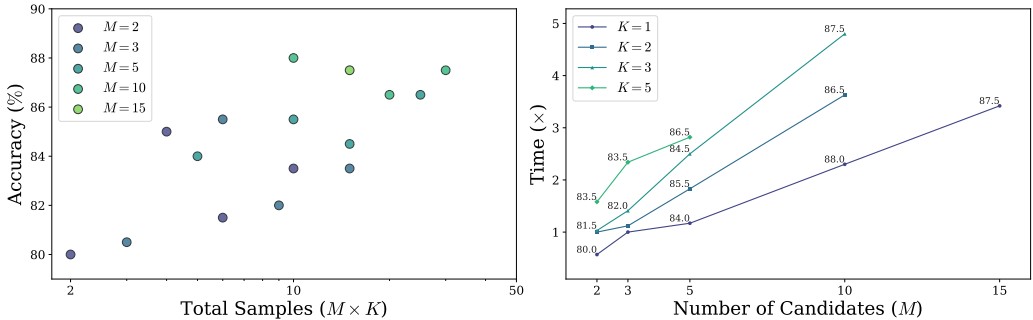

*Figure 7.* Effect of total samples on ETS. We ablate the hyerparameter settings with LLaDA-8B-Instruct and plot splited GSM8k accuracies (left) with corresponding latencies (right) for various sample counts (the accuracies are also reported).

## D.2. Reward Design

In Figure 9, we compares the correlations between the ground-truth reward and four self-evaluation reward candidates on GSM8K under Qwen3-1.7B model. As can be seen, only the self-consistency reward used in our ETS exhibits a clear separation between the two classes: answers that agree with the consensus receive consistently high reward, whereas incorrect answers receive low reward. In contrast, the distributions of the other three rewards largely overlap for correct and incorrect responses, meaning their numerical values do not reliably indicate semantic accuracy. This result confirms that

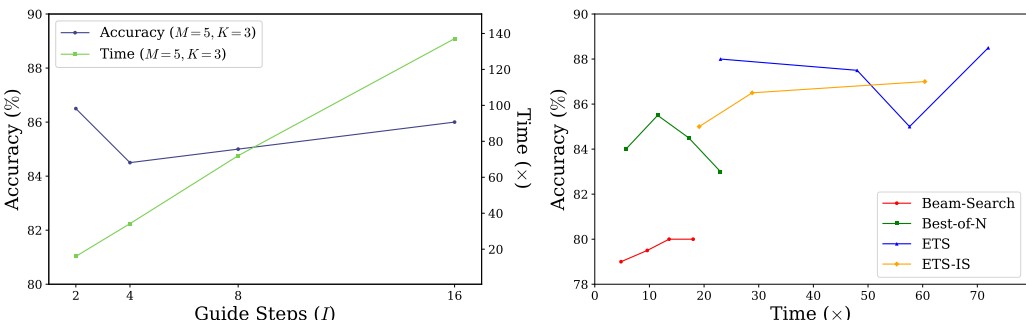

*Figure 8.* Left: Effect of guidance steps on ETS. We evaluate LLaDA-8B-Instruct on splited GSM8K with corresponding latencies are reported under various guidance steps. Right: Comparisons between TTS methods. We ablate the latencies under the LLaDA-8B-Instruct models evaluated on splited GSM8K for various TTS methods.

voting is uniquely effective as a self-evaluation signal for energy estimation in reasoning tasks.

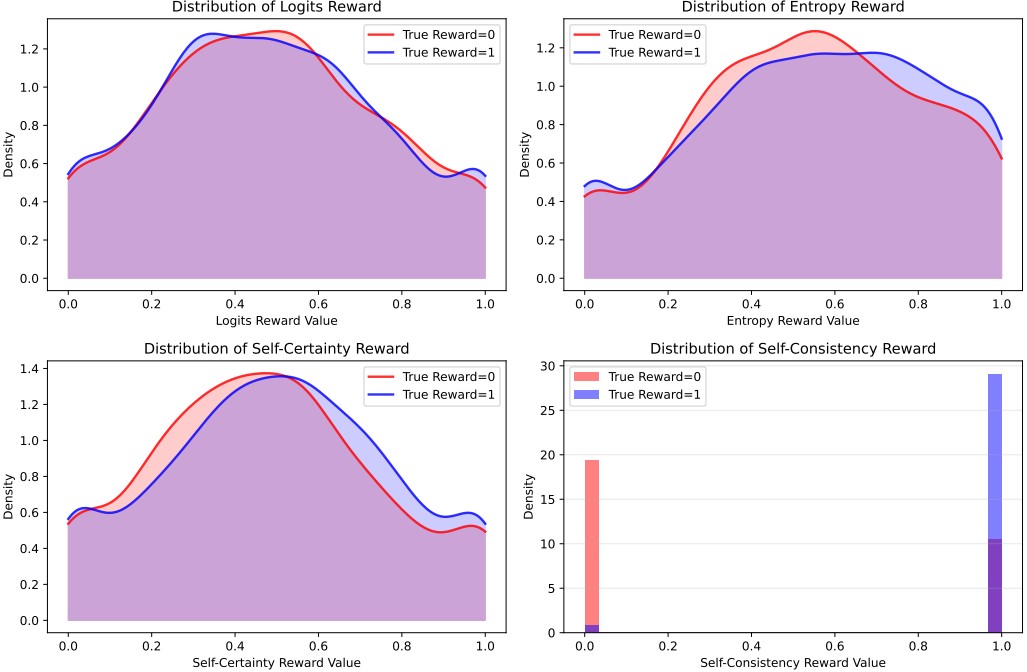

*Figure 9.* Comparison of self-evaluation rewards. We plot distributions of four rewards evaluated on GSM8K with Qwen3-1.7B (60% base accuracy), stratified by ground-truth reward (red: incorrect, blue: correct).

To further quantify the effectiveness of the self-consistency reward, Table 5 reports the performance of Qwen3-8B on GSM8K using both the self-consistency reward and the ground-truth reward. While the ground-truth reward naturally yields the best performance (e.g., pass@N accuracy of 98.10 for Best-of-N), our self-consistency reward achieves competitive results (94.09 for Best-of-N and 94.22 for ETS). This demonstrates that self-consistency serves as a practical, training-free proxy that closely approximates the true reward signal, validating its use in ETS when the ground-truth reward is unavailable during inference.

### D.3. Temperature

Following the settings in Appendix C.2, the temperature $t$ is adjusted based on the dataset and model characteristics: a lower temperature constrains sampling variability for models with lower base accuracy (or those that are only pre-trained), while a higher temperature promotes broader exploration for models with higher base accuracy (or post-trained models). Furthermore, a lower temperature encourages ARMs to early-stop more deterministically, thereby reducing the latency of

*Table 5.* Effect of reward on ETS. We evaluate Qwen3-8B on GSM8K under various rewards.

| Reward | Best-of-N | ETS | ETS-IS |
|---|---|---|---|
| Logits | 88.48 | 88.70 | 88.40 |
| Entropy | 90.30 | 89.54 | 88.93 |
| Self-Certainty | 90.07 | 89.46 | 88.93 |
| Self-Consistency | 94.09 | 94.22 | 91.81 |
| Ground-Truth | 98.10 (pass@N) | 97.57 | 95.00 |

ETS. However, a single temperature may not be optimal across all hyperparameter scales.

A notable exception is the GPQA dataset, where the verifier simply extracts the first capital letter from the response. As illustrated in Figure 10, this narrow criterion does not diminish the benefits of temperature tuning at high temperatures; instead, it leads to comparable performance between $t = 0.1$ and $t = 1.5$, indicating that even high temperatures can yield gains in this setting.

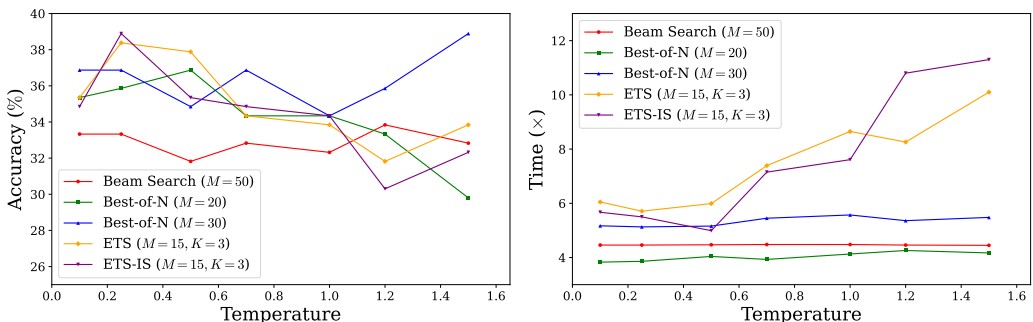

*Figure 10.* Effect of temperature on ETS. We ablate the temperature on Qwen3-8B and plot GPQA accuracies (left) with corresponding latencies (right).

Empirically, the optimal temperature is shared between Best-of-N and ETS with comparable latency (Chow et al., 2024), while Beam Search is insensitive to temperature (so we fix $t = 0.7$). Based on this, extensive experimental evidence suggests exploring $t$ within the range of $[0.25, 1.5]$.

### D.4. Generation Length

We ablate the maximum generation length $d_x$ for reasoning sequences. Results in Table 6 show that increasing $d_x$ beyond 512 tokens yields diminishing returns on most benchmarks (e.g., HumanEval). MATH500 is an exception, where longer sequences ($d_x = 1024$) are beneficial, due to their more complex reasoning chains. Based on this efficiency trade-off, we fix $d_x = 512$ for all main experiments on ARMs. For DLMs, we follow the original settings of LLaDA (in Table 4).

*Table 6.* Performance across generation lengths. We ablate the $d_x$ on Qwen3-8B and bold the best accuracy value for each method across different generation lengths.

| Length | MATH500 | | | | | | HumanEval | | | | | |
|---|---|---|---|---|---|---|---|---|---|---|---|---|
| | Base | Best-of-N | Beam Search | ETS | ETS-IS | GRPO | Base | Best-of-N | Beam Search | ETS | ETS-IS | GRPO |
| 256 | 47.2 | 49.8 | 44.6 | 55.6 | 49.4 | 56.0 | 60.98 | **67.07** | 67.68 | 68.90 | 62.80 | 62.80 |
| 512 | 60.0 | 65.2 | 61.2 | 72.4 | 66.2 | 69.2 | 58.54 | **67.07** | **69.51** | **71.34** | **68.9** | 65.85 |
| 1024 | 67.2 | **76.2** | **69.8** | **76.8** | **75.0** | **77.0** | **61.59** | 62.20 | 68.29 | 67.07 | 60.37 | **65.85** |

### D.5. Long-CoT Task

To address the concern that intrinsic self-consistency may be noisy in hard tasks, we have extended our evaluation to AIME24 (Zhang & Math-AI, 2024) on Qwen3-8B, a challenging long-CoT benchmark. Using the same reward design, we observe consistent improvements in accuracy, confirming that our method remains effective even when reasoning chains are long (6144 tokens) and diverse.

Furthermore, for long-CoT tasks, the high cost of $p_{ref}$ limits the sample budget of standard TTS methods. In contrast, ETS-IS achieves a large gain over ETS by exploiting $p_{small}$ to explore many more total samples within a comparable latency budget. This greater exploration yields substantially higher accuracy, as shown in Table 7.

*Table 7.* Performance on the AIME24 benchmark with Qwen3-8B (average over 32 samples), with best accuracy bolded.

| Method | AIME24 (avg@32) | Time (8×GPU) |
|---|---|---|
| Base | 16.25 | 1× |
| Beam Search | 16.46 | 1.64× |
| Best-of-N | 18.23 | 3.34× |
| ETS | 18.54 | 4.03× |
| ETS-IS | **22.40** | 2.82× |
| GRPO | 21.57 | 1× |

### D.6. ETS Asynchronous Pipeline

To fully exploit the parallel nature of ETS, we implement an asynchronous evaluation pipeline (Appendix B.2) that continuously dispatches generation tasks to GPU ranks, eliminating idle waiting periods common in synchronous evaluation loops (e.g., lm-eval).

As shown in Table 8, the pipeline yields different speedups depending on the inherent parallelism of each method. ETS and ETS-IS naturally generate many candidates and completions in parallel, allowing them to saturate the GPUs more effectively and thus obtain a larger relative speedup. In contrast, methods with lower parallelism (e.g., Base, Best-of-N) gain less acceleration. This asymmetry reflects algorithmic characteristics rather than unfairness. For a completely orthogonal assessment of algorithmic efficiency, we also report synchronous runtimes as a supplementary reference.

*Table 8.* Synchronous vs. asynchronous runtime on MATH500 with Qwen3-8B. The asynchronous pipeline is applied uniformly to all methods. Gains are computed as $(1 - \text{Async}/\text{Sync}) \times 100\%$.

| Method | Number of $x_0$ | Time (8×GPU) | | Gain |
|---|---|---|---|---|
| | | Sync | Async | |
| Base | 1 | 0:15:30 | 0:15:07 | 2.5% |
| Best-of-N | 50 | 0:40:23 | 0:37:26 | 7.5% |
| ETS | 115 | 1:32:05 | 1:02:32 | 32.6% |
| ETS-IS | 82.5 | 1:00:12 | 0:43:35 | 27.6% |

Accuracy remains unchanged between synchronous and asynchronous execution for all methods, confirming that the pipeline does not affect generation quality. The reported main results (Table 1) use the asynchronous pipeline for all TTS methods.

## E. Generation Example of ETS

Table 9 presents a step-by-step generation example of the ETS method applied to a GSM8K math problem by LLaDA-8B-Instruct. It illustrates the intermediate candidate completions, their corresponding energy scores, and the selection process across multiple guidance steps, culminating in the final answer. For every guidance step (1–6), the model produces five candidates (Candidate Index 1–5). For each candidate, a Monte Carlo procedure is applied to obtain three ($x\_0$) samples, from which final answers are extracted (shown in the "Extracted Answer" column). Taking guidance step 1 as an example, Candidate 1 yields the extracted answers (['\$90,0', '\$75,000', '\$90,000']), while Candidate 2 yields (['70000', '70000', '20000']), and so on for the remaining candidates. Within a given guidance step, the consensus answer is treated as the step-level target; here the consensus is '70000'. Rewards are then assigned by comparing each extracted answer to the consensus: answers matching the consensus receive reward 1, and mismatches receive reward 0. These rewards are aggregated into an energy that reflects the candidate's relative preference under the ETS selection rule. Consequently, Candidate 1 in Step 1 receives rewards ([0,0,0]) and has Energy (=1), whereas Candidate 2 receives rewards ([1,1,0]) and attains a substantially stronger selection score (Energy (=14684.64) as reported). The "Selected Candidate" column records the candidate chosen for that step (Candidate 2 for Step 1), and the same consensus–reward–energy–selection mechanism is repeated identically for subsequent steps until the final solution is formed, culminating in the stated final answer 70000.

*Table 9.* Step-by-step generation example of ETS on GSM8K using LLaDA-8B-Instruct. The table illustrates the intermediate candidate texts, their energy scores, and the selection process across multiple guidance steps ($I = 8$) for a math word problem. The response's meaningful content is completed at guidance step 6, and subsequent steps produce only the `<|endoftext|>` token, which is omitted for space consideration. (Part 1 of 2)

| Step | Candidate Index | Generated Text | Extracted Answer | Energy | Selected Candidate |
|---|---|---|---|---|---|
| | | Question: Josh decides to try flipping a house. He buys a house for $80,000 and then puts in $50,000 in repairs. This increased the value of the house by 150%. How much profit did he make?\nAnswer: | | | |
| 1 | 1 | To determine Josh's profit, we need to follow these steps:\n\n1. Calculate the increased value of the house after the repairs.\n2. Calculate | ['$90,0', '$75,000', '$90,000'] | 1 | 2 |
| | 2 | Josh bought the house for $80,000 and put $50,000 in repairs, so his total cost was $80 | ['70000', '70000', '20000'] | 14684.64 | |
| | 3 | To determine Josh's profit, we need to follow these steps:\n\n1. Calculate the increased value of the house after the repairs.\n2. Sub | ['120,000', '45,000', '$120, 000'] | 1 | |
| | 4 | To determine the profit Josh made, we need to follow these steps:\n\n1. Calculate the increased value of the house after the repairs.\n2. | ['120,000', '45,000', '$120,000'] | 1 | |
| | 5 | Josh bought the house for $80,000 and spent $50,000 on repairs, so his total cost was $80 | ['5000', '195000', '70000'] | 7342.82 | |
| 2 | 1 | ,000 + $50,000 = $130,000.\nThe repairs increased the value of the house by | ['70000', '70000', '70000'] | 22026.47 | 1 |
| | 2 | ,000 + $50,000 = $130,000.\nThe repairs increased the value of the house by | ['10000', '70000', '65000'] | 7342.82 | |
| | 3 | ,000 + $50,000 = $130,000.\nThe repairs increased the value of the house by | ['70000', '70000', '70000'] | 22026.47 | |
| | 4 | ,000 + $50,000 = $130,000.\nHe sold the house for a value that increased | ['10000', '195000', '195000'] | 1 | |
| | 5 | ,000 + $50,000 = $130,000.\nThe repairs increased the value of the house by | ['70,000', '70000', '5000'] | 7342.82 | |
| 3 | 1 | 150%, so the increased value is 150% of $80,000, which is 1.5 * $ | ['70000', '$70,000', '70000'] | 14684.64 | 4 |
| | 2 | 150%, so the new value of the house was $80,000 * 1.5 = $120,0 | ['10000', '20000', '10000'] | 1 | |
| | 3 | 150%, so the new value of the house is $80,000 * 1.5 = $120,0 | ['100000', '2,000', '50000'] | 1 | |
| | 4 | 150%, so the house is now worth $80,000 + (150/100) * $80 | ['70000', '70000', '70000'] | 22026.47 | |
| | 5 | 150%, so the new value of the house is $80,000 * 150/100 = $1 | ['10000', '70000', '5000'] | 7342.82 | |

*Table 9.* (**Continued**) Step-by-step generation example of ETS on GSM8K using LLaDA-8B-Instruct. (Part 2 of 2)

| Step | Candidate Index | Generated Text | Extracted Answer | Energy | Selected Candidate |
|------|------|------|------|------|------|
| 4 | 1 | ,000 = \$80,000 + \$120,000 = \$200,000.\n | ['70000', '70000', '70000'] | 22026.47 | 5 |
| | 2 | ,000 = \$80,000 + \$120,000 = \$200,000.\n | ['70000', '70000', '70000'] | 22026.47 | |
| | 3 | ,000 = \$80,000 + \$120,000 = \$200,000.\n | ['70000', '70000', '70000'] | 22026.47 | |
| | 4 | ,000 = \$80,000 + \$120,000 = \$200,000.\n | ['70000', '70000', '70000'] | 22026.47 | |
| | 5 | ,000 = \$80,000 + \$120,000 = \$200,000.\n | ['70000', '70000', '70000'] | 22026.47 | |
| 5 | 1 | He sold the house for \$200,000, so his profit is \$200,000 - \$130 | ['70000', '70000', '70000'] | 22026.47 | 4 |
| | 2 | To find the profit, we subtract the total cost from the new value of the house: \$200,000 - \$130, | ['70000', '70000', '70000'] | 22026.47 | |
| | 3 | To find the profit, we subtract his total cost from the new value of the house: \$200,000 - \$130, | ['70000', '70000', '70000'] | 22026.47 | |
| | 4 | To find the profit, we subtract the total cost from the new value of the house: \$200,000 - \$130, | ['70000', '70000', '70000'] | 22026.47 | |
| | 5 | To find the profit, we subtract the total cost from the new value of the house: \$200,000 - \$130, | ['70000', '70000', '70000'] | 22026.47 | |
| 6 | 1 | ,000 = \$70,000.\n#### 70000 | ['70000', '70000', '70000'] | 22026.47 | 2 |
| | 2 | ,000 = \$70,000.\n#### 70000 | ['70000', '70000', '70000'] | 22026.47 | |
| | 3 | ,000 = \$70,000.\n#### 70000 | ['70000', '70000', '70000'] | 22026.47 | |
| | 4 | ,000 = \$70,000.\n#### 70000 | ['70000', '70000', '70000'] | 22026.47 | |
| | 5 | ,000 = \$70,000.\n#### 70000 | ['70000', '70000', '70000'] | 22026.47 | |

Final Answer: Josh bought the house for \$80,000 and put \$50,000 in repairs, so his total cost was \$80,000 + \$50,000 = \$130,000.\nThe repairs increased the value of the house by 150%, so the house is now worth \$80,000 + (150/100) * \$80,000 = \$80,000 + \$120,000 = \$200,000.\nHe sold the house for \$200,000, so his profit is \$200,000 - \$130,000 = \$70,000.\n#### 70000

