# OpenReview forum: "ETS: Energy-Guided Test-Time Scaling for Training-Free RL Alignment"
_ICML.cc/2026/Conference — ICML 2026 regular_

### Official Review · Reviewer_CFo5 · 2026-03-04

**Soundness:** 3
**Presentation:** 3
**Significance:** 4
**Originality:** 4
**Overall Recommendation:** 4
**Confidence:** 3

**Summary:**

This submission proposes a training-free inference method that allows to sample directly from the optimal Reinforcement Learning policy, and an Energy-Guided Test-Time Scaling algorithm that uses this to estimate the main energy term using a Monte Carlo approach.

They test the method with different approaches such as autoregressive models and diffusion language models on a variety of benchmarks (coding, reasoning, science benchmarks) and show improved generation quality.

**Compliance With Llm Reviewing Policy:**

Affirmed.

**Final Justification:**

Given that the authors have addressed my concerns and clarified the applicability of their method in practical settings, I have raised my score as well.

**Key Questions For Authors:**

- If the optimal RL distribution is known in closed form, can we sample from it directly at inference time, without any additional training?  i.e. it's unclear to me how this approach would be applicable in real-world applications as opposed to theory

 - "by replacing samples from pref with those from a computationally cheaper proposal model, we can significantly reduce the latency while preserving the correctness of the estimator." -- how do you choose the model for sampling from it?

- Have you tried models other than Qwen3 for ARMs?

**Limitations:**

The impact statement of this work is practically nonexistent.
"there are many potential societal consequences of our work, none which we feel must be specifically highlighted here" -- why is this the case? Surely you can think of some potential ways in which your approach can have consequences when deployed in the real world?

**Strengths And Weaknesses:**

Strengths:
- Reducing the computational cost (and therefore monetary and energy cost) of deployment is an important problem given the ubiquity of ML today
- The method proves better accuracy while having comparable latency to other TTS methods.
- The approach is theoretically well-grounded and explained


Weaknesses:
- As someone with limited experience in RL, it is hard to read the paper because of missing acronym explanations, e.g. RLHF,  PPO, DPO, and GRPO, and there are assumptions of background knowledge that the authors build upon that are not explicitly stated
- The related work section seems more like a review, without positioning how the current approach compares (at a high-level) to previous ones
- I'm unclear how this approach would work in real-life applications, when the optimal distribution is unknown

---

> ### Author Rebuttal · Authors · 2026-03-31
>
> Thanks for your review. Here are the replies.
>
> > Weakness 1: As someone with limited experience in RL, it is hard to read the paper because of missing acronym explanations, e.g. RLHF, PPO, DPO, and GRPO, and there are assumptions of background knowledge that the authors build upon that are not explicitly stated.
>
> Answer: Thank you for pointing this out. We will add brief explanations when first introducing RLHF, PPO, DPO, and GRPO to improve accessibility for readers less familiar with RL literature.
>
> > Weakness 2: The related work section seems more like a review, without positioning how the current approach compares (at a high-level) to previous ones.
>
> Answer: Our related work section begins by positioning our method within the Test‑Time Scaling (TTS) literature, and we explicitly highlight distinctions from prior work **in line 108**: "unlike Power Sampling and Quest, our method avoids slow Metropolis‑Hastings sampling"; **and in line 155 (Remark 1) **compared to recent diffusion‑based guided generation, our approach applies to both ARMs and DLMs, offers theoretical convergence guarantees, and achieves higher efficiency through our acceleration techniques." We will clarify this more clear in the revised version.
>
> > Weakness 3 & Question 1: I'm unclear how this approach would work in real-life applications, when the optimal distribution is unknown ... can we sample from it directly at inference time ....
>
> Answer: Yes, as you suggested, the core idea of our method is **directly sample from the optimal distribution** without requiring the explicit formulation of optimal distribution. Our Proposition 2 and Algorithm 1 energy‑based guidance method of sampling from optimal distribution, making it readily applicable to real‑world scenarios.
>
> > Question 2: "by replacing samples from pref with those from a computationally cheaper proposal model, we can significantly reduce the latency while preserving the correctness of the estimator." -- how do you choose the model for sampling from it?
>
> Answer: As clarified in Section 5.2, 5.3.
>
> For dLLM, we use a lossless acceleration frameworks Fast-dLLM, which that achieve speedup without efforts.
>
> For ARMs, as discussed in Section 5.2, there is no appropriate lossless acceleration framework, e.g., EAGLE will oppositely slow down sampling (see W5 to Reviewer Kp4R for details). Thus, we use Qwen3‑1.7B as the small model (distilled from Qwen3‑8B), which is efficient and accurate in practice (Table 1 and Figure 5).
>
>
> > Question 3: Have you tried models other than Qwen3 for ARMs?
>
> Answer: Thanks for your suggestion, we extend our methods to Llama‑3.1‑8B‑Instruct and confirm that ETS works consistently. We primarily chose Qwen3 because its lightweight variants benefit from stronger distillation and are better supported by existing acceleration frameworks, making them particularly suitable for our ETS-IS setup.
>
> Nonetheless, the results on Llama‑3.1‑8B‑Instruct demonstrate that ETS generalizes well beyond the Qwen family. We are unable to evaluate ETS‑IS on Llama‑3.1 because the model family does not include a suitably small distilled variant for importance sampling.
>
> |Method|MATH500|Time|
> |:-|:-:|:-:|
> |Base|34.2|1x|
> |Beam Search|34.0|1.28x|
> |Best-of-N|49.8|2.63x|
> |ETS|51.4|3.54x|
>
> > Limitations: The impact statement of this work is practically nonexistent. ...
>
> Answer: Thanks for this important comment. Our approach lowers the computational cost of inference-time alignment, making high-quality AI more accessible and energy-efficient. This can democratize alignment techniques for resource-constrained settings and reduce the environmental footprint of large language model deployment. We will update the impact statement to reflect these positive societal benefits.
>
> **We're happy to address any further questions you may have.**

---

> > ### Author Rebuttal · Reviewer_CFo5 · 2026-04-02
> >
> > The authors adequately addressed my comments, and clarified any points that were unclear. I will raise my score.

---

> > > ### Author Response · Authors · 2026-04-03
> > >
> > > Thank you for your valuable comments and engaging discussion. We appreciate the time and insight you've provided to strengthen our manuscript.
> > >
> > > We are pleased to have responded to your points and addressed your concerns. We also note your earlier comment that you would raise your score, and we look forward to your final evaluation. We will carefully incorporate your feedback and suggestions into the manuscript during revision.
> > >
> > > We remain available until the end of the discussion phase for any further questions or clarifications needed.
> > >
> > > Sincerely,
> > > The Authors

---

### Official Review · Reviewer_Dtkx · 2026-03-11

**Soundness:** 2
**Presentation:** 3
**Significance:** 3
**Originality:** 3
**Overall Recommendation:** 4
**Confidence:** 2

**Summary:**

The motivation for this paper is that RL post-training for LLM alignment is costly and unstable. This paper proposes ETS, a training-free inference method that samples directly from the optimal RL policy. Main insight: under a unified MLM framework covering both autoregressive models and diffusion language models, the optimal transition kernel decomposes into a reference policy term and an energy term, which ETS estimates via online Monte Carlo with a provable convergence rate. They perform experiments on reasoning, coding, and science benchmarks.

**Compliance With Llm Reviewing Policy:**

Affirmed.

**Final Justification:**

I'd like to keep my original score (4). In the rebuttal, the authors added AIME24 results with avg@32, but this isn't quite the same as a proper statistical significance testing (e.g., paired permutation tests, etc). The original concerns about small gaps in the results reported in the paper remained unaddressed.

**Key Questions For Authors:**

1. The paper lacks statistical significance tests. Some differences between methods are quite small, so it's hard to be confident these gaps are meaningful rather than noise. Can you do this?

2. All benchmarks are verifiable tasks (math, coding, multiple-choice), which is where self-consistency works best. How well would this reward signal work in more open-ended tasks where multiple valid answers exist?

3. The experiments are limited to relatively small models (1.7B and 8B). Do you expect ETS to provide similar gains on larger models that already have higher accuracy?

4. The DLM gains seem weaker than the ARM gains across the board, and the latency is much higher. Do you have insights on why this is the case?

**Strengths And Weaknesses:**

Soundness: The theoretical framework is clean. The convergence bound in total variation (Proposition 3) is a nice contribution. The empirical results are generally convincing within the evaluated setup. They frame the paper as training-free inference (as a way to replace RL post-training), and the results make a reasonable case for verifiable reasoning tasks. That said, I'm not sure if the comparison with GRPO is fair since their method uses more compute at inference time, while GRPO pays that cost at training time. The comparison with other inference-time methods is fair, though. Also, the paper is missing statistical significance tests.

Presentation: The paper is well-written and the narrative is easy to follow. Figure 2 gives a clear overview. I like the unified MLM framing even though in practice the acceleration strategies for ARMs and DLMs are different. I'm not sure I'd average the “time” in Table 1.

Significance: The problem is very relevant and the fact that it does not require training nor human preference data is appealing. The experimental results within the evaluated scope seem good to me, but it would be good if the scope was broader (see my points below).

Originality: The idea is novel to my knowledge, and the paper clearly positions itself relative to prior work.

---

> ### Author Rebuttal · Authors · 2026-03-31
>
> Thanks for your review. Here are the replies.
>
> > Presentations: ... I'm not sure I'd average the “time” in Table 1.
>
> Answer: The "Time" in Table 1 is averaged over all data, giving a representative measure of average latency across tasks.
>
> > Weakness 1: That said, I'm not sure if the comparison with GRPO is fair since their method uses more compute at inference time, while GRPO pays that cost at training time.
>
> Answer: As you said, our comparison to TTS methods are fair, and our method is improves them in both effect and effectiency.  For GRPO, it is not viewed as a compute-matched baseline, but to demonstrate that our inference-time method achieves alignment of comparable quality to a training-based RL approach. This follows the standard practice in inference-time alignment (e.g., power sampling [1]), where training-time cost is traded for inference-time compute.
>
> > Weakness 2 & Question 1: The paper lacks statistical significance tests. Some differences between methods are quite small, so it's hard to be confident these gaps are meaningful rather than noise.
>
> Answer: Thanks for pointing out this, To address your concern, we implement our method on AIME24 with  statistical mean accuracy avg@32 reported, (on Qwen3-8B). The following table confirms that our improvements are statistically meaningful.
>
> |Method|AIME24 (avg@32)|Time|
> |:-|:-:|:-:|
> |Base|16.25|1x|
> |Beam Search|16.46|1.64x|
> |Best-of-N|18.23|3.34x|
> |ETS|18.54|4.03x|
> |ETS-IS|22.40|2.82x|
> |GRPO|21.57|1x|
>
> > Question 2: ... How well would this reward signal work in more open-ended tasks where multiple valid answers exist?
>
> Answer: To evaluate generalization beyond verifiable tasks, we extended our experiments to open-ended task IFEval with Qwen3-8B. Since many open‑ended tasks **do not yield extractable answers** for self‑consistency, we instead use self‑certainty as the reward signal, which might  weaken our method as discussed in Appendix D.2. Fortunately, the following results show that our method has though slightly lower, but significant improvements under open-ended task.
>
> |Method|IFEval|Time|
> |:-|:-:|:-:|
> |Base|31.61|1x|
> |Best-of-N|31.42|3.02x|
> |ETS|32.90|6.59x|
> |ETS-IS|32.35|5.87x|
>
> > Question 3: ... Do you expect ETS to provide similar gains on larger models that already have higher accuracy?
>
> Answer: As your suggested, we implement our method on larger Qwen3‑30B‑A3B, showing that ETS continues to deliver meaningful gains even for larger, higher‑accuracy models.
>
> Moreover, our results on AIME24 and Llama (Q3 to Reviewer CFo5) further verify the improvement of our method can be extended to harder task and more LLM structures.
>
> |Method (Qwen3‑30B‑A3B)|MATH500|Time|
> |:-|:-:|:-:|
> |Base|63.0|1x|
> |Beam Search|63.6|3.70x|
> |Best-of-N|71.2|9.69x|
> |ETS|71.6|11.9x|
> |ETS-IS|69.2|3.23x|
> |GRPO|69.9|1x|
>
> |Method (Llama-3.1-8B-Instruct)|MATH500|Time|
> |:-|:-:|:-:|
> |Base|34.2|1x|
> |Beam Search|34.0|1.28x|
> |Best-of-N|49.8|2.63x|
> |ETS|51.4|3.54x|
>
> >Question 4: The DLM gains seem weaker than the ARM gains across the board, and the latency is much higher. Do you have insights on why this is the case?
>
> Answer: Performance gains depend on base model accuracy—DLMs have lower base accuracy than ARMs, leaving less room for improvement. Additionally, DLM inference infrastructures are less optimized than ARM implementations (e.g., without KVcache), leading to higher latency across all TTS methods (e.g., Best-of-N).
>
> **We're happy to address any further questions you may have.**
>
> Reference
>
> [1] A. Karan and Y. Du. Reasoning with Sampling: Your Base Model is Smarter Than You Think. ICLR 2026.

---

> > ### Author Rebuttal · Reviewer_Dtkx · 2026-03-31
> >
> > - On statistical significance tests: you added AIME24 results with avg@32, but this isn't quite the same as a proper statistical significance testing (e.g., paired permutation tests, etc). The original concerns about small gaps in the results reported in the paper remain unaddressed
> > - On open-ended tasks: you mentioned "but significant improvements under open-ended task"; again, significant in which sense? see my comment above
> > - On larger models: thanks for the update
> > - On DLMs: you mentioned "DLMs have lower base accuracy than ARMs, leaving less room for improvement."; isn't this counterintuitive?

---

> > > ### Author Response · Authors · 2026-04-03
> > >
> > > Thanks for your feedback. Here are the replies.
> > >
> > > > Reason 1 & Reason 2: The original concerns about small gaps in the results reported in the paper remain unaddressed.
> > >
> > > Answer: To directly address the concern about statistical significance, we performed independent two‑sample t‑tests (equal variance assumed) on five independent runs for each method. The results are summarized below:
> > >
> > > |Comparison (MATH500)|Mean Difference|t‑statistic|df|p‑value (two‑tailed)|
> > > |:-|:-:|:-:|:-:|:-:|
> > > |Base vs. Best‑of‑N|-5.40|11.13|8|<0.001|
> > > |Base vs. ETS|-10.28|15.94|8|<0.001|
> > > |Best‑of‑N vs. ETS|-4.88|6.87|8|<0.001|
> > >
> > > |Comparison (IFEval)|Mean Difference|t‑statistic|df|p‑value (two‑tailed)|
> > > |:-|:-:|:-:|:-:|:-:|
> > > |Base vs. Best‑of‑N|-2.55|3.26|4.5|<0.05|
> > > |Base vs. ETS|-4.03|3.99|8|<0.01|
> > > |Best‑of‑N vs. ETS|-1.48|2.14|4.6|<0.05|
> > >
> > > > Reason 4: you mentioned "DLMs have lower base accuracy than ARMs, leaving less room for improvement."; isn't this counterintuitive?
> > >
> > > Answer: The statement that lower base accuracy leaves less room for improvement refers to **absolute gains, not relative improvements**. In both our experiments (Table 1) and other DLM studies [1], DLMs consistently exhibit lower absolute accuracy than ARMs of comparable scale.
> > >
> > > |Models|MATH500|GSM8K|GPQA-Diamond|HumanEval|
> > > |:-|:-:|:-:|:-:|:-:|
> > > |LLaDA-8B-Instruct|42.2|80.36|28.28|35.98|
> > > |Qwen2.5-7B|49.8|85.4|31.31|57.9|
> > > |Qwen3-8B|60.0|89.39|34.34|58.54|
> > >
> > > Compared to ARMs of similar scale, **the lower base accuracy and weaker reasoning capabilities of DLM inherently limits the effectiveness of Test‑Time Scaling methods**. Even so, ETS still achieves substantial relative improvements on DLMs, demonstrating its robustness and generalization.
> > >
> > > |Models|Method|Average|Improvement|
> > > |:-|:-|:-:|:-:|
> > > |Qwen3-1.7B|Base|43.98|-|
> > > |Qwen3-1.7B|ETS|54.76|+24.5%|
> > > |Qwen3-8B|Base|60.57|-|
> > > |Qwen3-8B|ETS|69.34|+14.5%|
> > > |LLaDA-8B-Instruct|Base|46.71|-|
> > > |LLaDA-8B-Instruct|ETS|51.01|+9.2%|
> > >
> > > Reference
> > >
> > > [1] S. Nie et al. Large Language Diffusion Models. 2025.
> > >
> > > **We're happy to address any further questions you may have.**

---

### Official Review · Reviewer_Kp4R · 2026-03-12

**Soundness:** 2
**Presentation:** 2
**Significance:** 2
**Originality:** 2
**Overall Recommendation:** 3
**Confidence:** 3

**Summary:**

This paper proposes Energy-Guided Test-Time Scaling (ETS), a training-free inference method that enables direct sampling from the optimal RL policy for language model alignment. The key insight is that the KL-regularized RL objective admits a closed-form solution where the optimal backward transition kernel decomposes into (1) the reference model's transition and (2) an energy term representing the conditional expectation of exponentiated rewards. ETS estimates this energy term via online Monte Carlo sampling with importance sampling acceleration, providing theoretical convergence guarantees. The method applies to both autoregressive models (ARMs) and diffusion language models (DLMs) under a unified masked language modeling (MLM) framework.

**Compliance With Llm Reviewing Policy:**

Affirmed.

**Final Justification:**

Slightly raise my score considering that most technical issues have been addressed in the rebuttal, however, I still think that the write-up could be largely improved for better readability

**Key Questions For Authors:**

While training-free approaches are important, I would suggest designing methods that do not rely on too many approximations.

**Strengths And Weaknesses:**

Pros:
1. The motivation is clear.
2. ETS handles both ARMs and DLMs under a single MLM formulation, demonstrating broad applicability across different generative paradigms.
3. The paper presents loads of theoretical justifications.
4. The performance seems to be good especially given that the proposed method is training-free.

Cons:
1. The writing can be polished, especially the usage of symbols. I would suggest swapping the roles of x and y and also why we should generate x from x_T to x_0, making the reading comprehension more challenging.
2. The proposed method requires a preset T which indicated the length of the output, for ARMs, this setup is unreasonable.
3. The proposed method suffers from high computational overhead and the generation process is also a bit complicated. The method first samples M candidates and then for each candidate applies MCTS to generate K commpletions, which is pretty time-consuming. Also, the estimation can suffer from high variance or bias when the generation is pretty long, given that deep reasoning is pretty important for recent LLMs.
4. The reward estimation is also not accurate enough in many cases especially for hard tasks.
5. While the paper proposes an acceleration method using importance sampling, the distribution between the small model and the ref model can be huge.

---

> ### Author Rebuttal · Authors · 2026-03-31
>
> Thanks for your review. Here are the replies.
>
> > W1: The writing can be polished, especially the usage of symbols $x_{0}$ and $x_{T}$...
>
> Answer: We adopt this notation to align with the standard formulation in Mask Language Model, where generation proceeds from a fully masked state $x_T$ to the clean output $x_0$ [1]. This is a widely accepted convention in the literature.
>
> > W2: The method requires an unreasonable preset T for ARM...
>
> Answer: For ARMs, generation naturally requires a maximum length bound T (e.g., the `max_new_tokens` argument in Hugging Face’s `generate()` [2]). Setting T is a standard practice in inference-time algorithms and does not introduce additional assumptions beyond what is already used in baseline methods.
>
> > W3.1: The method suffers from high computational overhead and a bit complicated generation over $M$ and $K$ MCTS...
>
> Answer: We clarify that our method **does not use MCTS**; it simply performs Monte Carlo estimation with M candidates and K completions per candidate. Besides, as clarified in Algorithm 1, our guidance is only applied at the block level rather than per token,  making the overhead modest. Here is an example in a guidance step (See Figure 2) with Qwen3-8B.
>
> |Method|Sample M candidates|Sample M*K completions|Time|MATH500|
> |:-|:-:|:-:|:-:|:-:|
> |ETS|17.8s (M=10)|42.5s (K=3)|60.3s|72.4|
> |ETS-IS|17.8s (M=10)|23.8s (K=3)|41.6s|66.0|
> |Best-of-N|43.2s (M=50)|-|43.2s|65.2|
>
> The results together with our Table 1 and Figure 5 indicates our improvement in effciency and effect over TTS methods.
>
> Moreover, we observe that the latency is largely bounded by hardware rather than algorithm complexity. On higher‑performance GPUs, together with our asynchronous pipeline in Appendix B.2, our methods are more efficient, as shown in the table below.
>
> |Method|MATH-500|Time (4xH20-3e, original)|Time (8xH200)|Relative Acceleration|
> |:-|:-:|:-:|:-:|:-:|
> |Base|60.0|0:26:55 (1x)|0:15:07 (1x)|-|
> |ETS|72.4|4:26:43 (9.85x)|1:02:32 (4.13x)|2.39x|
> |ETS-IS|66.2|2:38:15 (5.88x)|0:43:35 (2.89x)|2.03x|
>
> > W3.2: The estimation of reward can suffer from high variance or bias when the generation is pretty long ...
>
> Answer: Our following results on hard AIME24 (with reasoning chains up to 6144 tokens) indicate the reward estimation converges quickly with increasing K so that modest K is already suffices, confirming our stable estimation even in long‑CoT settings. Besides that, as clarified in W1 to Reviewer 7tTM, our reward is voted based on all existing trajectory, further reducing its intristic bias.
>
> |Reward|K=2|K=3|K=5|K=10|
> |:-|:-:|:-:|:-:|:-:|
> |Ground-Truth|0.13|0.171|0.157|0.165|
> |Self-Consistency|0.167|0.209|0.181|0.176|
>
> > W4: The reward estimation is also not accurate enough in many cases especially for hard tasks.
>
> Answer: To address the concern that self-consistency may be not accurate enough in hard tasks, we have extended our evaluation to AIME24 on Qwen3-8B. Using the same reward design, we observe consistent improvements in accuracy, confirming that our method remains effective even when reasoning chains are long and diverse.
>
> |Method|AIME24 (avg@32)|Time|
> |:-|:-:|:-:|
> |Base|16.25|1x|
> |Beam Search|16.46|1.64x|
> |Best-of-N|18.23|3.34x|
> |ETS|18.54|4.03x|
> |ETS-IS|22.40|2.82x|
> |GRPO|21.57|1x|
>
> Besides, our results under Llama in Q3 to Reviewer CFo5 further verifies the generalization of our method.
>
> > W5: Under importance sampling, the distribution between the small model and the ref model can be huge.
>
> Answer: For dLLM, we use a lossless acceleration frameworks that **achieve speedup without any distributional shift**. For ARM, we select the proposal small model as a distilled version of the reference model to minimize distributional shift (Section 5.2), so that in most cases e.g., GPQA, HumanEval, and AIME24, the reward under importance sampling is accurate.
>
> As clarified in Section 5.2, there is no appropriate lossless acceleration method for ARM, The existing methods e.g., EAGLE‑3 [4] are limited to batch size 1, **inducing additional latency in practice** compared to batching framework. If a batched version of EAGLE is available, our method can achieve even higher throughput.
>
> |Model|Method|p_small|GSM8K|Time|
> |:-|:-|:-|:-:|:-:|
> |Qwen3-1.7B|ETS (bsz=M*K)|-|81.88|8.94x|
> |Qwen3-1.7B|ETS-IS (bsz=1)|Qwen3-1.7B_eagle3|82.58|13.6x|
> |Qwen3-8B|ETS (bsz=M*K)|-|94.24|6.21x|
> |Qwen3-8B|ETS-IS (bsz=M*K)|Qwen3-1.7B|91.81|5.33x|
> |Qwen3-8B|ETS-IS (bsz=1)|Qwen3-8B_eagle3|94.70|9.39x|
>
> >Q1: While training-free approaches are important, I would suggest designing methods that do not rely on too many approximations.
>
> Answer: Our approach can be categorized into TTS. It can be viewed as directly implementing the optimal energy‑based distribution using a small proposal model, which is a principled and widely adopted technique.
>
> Reference
>
> [1] J. Ou et al. Your Absorbing Discrete Diffusion Secretly Models the Conditional Distributions of Clean Data. ICLR 2025. [2] https://huggingface.co/blog/how-to-generate

---

> > ### Author Rebuttal · Reviewer_Kp4R · 2026-04-04
> >
> > Thanks for the reply, however, autoregressive decoding rather than masked language modeling is the mainstream. Besides, for ARM, a special token (e.g., eos) would be used to terminate the generation process adaptively.

---

> > > ### Author Response · Authors · 2026-04-04
> > >
> > > Thanks for your feedback. Here are the replies.
> > >
> > > > Reason 1: however, autoregressive decoding rather than masked language modeling is the mainstream.
> > >
> > > Answer: While autoregressive decoding is indeed the mainstream paradigm, a key advantage of our work is **the unified framework that supports both ARMs and DLMs**, offering flexibility across different use cases. The MLM notation is adopted precisely to enable this unification, not to replace the AR convention.
> > >
> > > > Reason 2: Besides, for ARM, a special token (e.g., eos) would be used to terminate the generation process adaptively.
> > >
> > > Answer: We would like to clarify that both our ARM and DLM implementations **follow standard generation logic in their respective paradigms**. For ARMs, we use the standard decoding practice: a maximum length T is pre‑set (as in any generation API or `generate()` function), but generation stops immediately upon emitting the EOS token, so adaptive termination is fully preserved.
> > >
> > > **We're happy to address any further questions you may have.**

---

### Official Review · Reviewer_7tTM · 2026-03-13

**Soundness:** 3
**Presentation:** 3
**Significance:** 2
**Originality:** 2
**Overall Recommendation:** 4
**Confidence:** 3

**Summary:**

This work proposes a novel algorithm for Test-Time Scaling (TTS), addressing a critical challenge in enhancing model performance during inference. A key strength of the proposed method is its versatility, as it is effectively applicable to both Autoregressive Language Models (ALMs) and Diffusion Language Models (DLMs). Experimental results demonstrate that this approach achieves significant performance improvements across diverse benchmarks while maintaining relatively low computational costs.

**Compliance With Llm Reviewing Policy:**

Affirmed.

**Final Justification:**

The remaining concerns are not addressed. I keep my evaluation unchanged.

**Key Questions For Authors:**

Please refer to the weaknesses outlined above.

**Limitations:**

Please refer to the weaknesses-1 outlined above.

**Strengths And Weaknesses:**

## Strengths
1. **Broad Model Compatibility**: A significant strength of this work is the versatility of the proposed Energy-Guided Test-Time Scaling (ETS) algorithm. Unlike many existing methods tailored to specific architectures, ETS demonstrates robust effectiveness across both Autoregressive Language Models (ALMs) and Diffusion Language Models (DLMs), highlighting its potential as a unified framework for test-time compute allocation.

2. **Efficient Performance Gains**: The method successfully strikes a balance between efficacy and efficiency. Experimental results indicate that ETS achieves substantial improvements in generation quality across reasoning, coding, and science benchmarks while maintaining relatively low computational overhead compared to baseline scaling strategies.

## Weaknesses
1. **Limitations of Reward Design and Intrinsic Consistency**:
Section 4.4 details the reward design, which relies on intrinsic consistency as a primary metric. While practical, this approach has inherent limitations (e.g., it may fail to capture factual correctness if the model consistently hallucinates, or it might penalize diverse but valid reasoning paths).
- The paper lacks a critical discussion on how sensitive the final performance is to the specific formulation of this consistency metric.
- It remains unclear whether the observed gains are robust across different types of tasks where intrinsic consistency might be a noisy or misleading signal. A deeper analysis or ablation study on the impact of alternative reward signals would strengthen the claims.

2. **Insufficient Ablation on the Accelerated Evaluation Protocol**:
In Appendix B.2, the authors introduce a specialized evaluation protocol (scheduling process) designed to accelerate the inference speed. However, the main experimental results rely on this optimized setup without quantifying its specific contribution to the reported performance.
- *Missing Baseline Comparison*: It is unclear whether the reported efficiency gains stem from the core ETS algorithm itself or largely from this custom scheduling mechanism.
- *Necessity of the Protocol*: Would the method still outperform baselines if standard evaluation procedures were used? If removing this protocol leads to a significant drop in throughput or even accuracy, this trade-off should be explicitly discussed and quantified in the main text to provide a fair assessment of the algorithm's standalone utility.

---

> ### Author Rebuttal · Authors · 2026-03-31
>
> Thanks for your review. Here are the replies.
>
> > Weakness 1: Limitations of Reward Design and Intrinsic Consistency: ... While practical, this approach has inherent limitations ...
>
> Answer: The reliance on intrinsic consistency is indeed a known challenge in unsupervised RLVR [1]. However, rather than a single consistency judgment, our approach mitigates this by aggregating votes **across the all trajectories** of sampled outputs, which significantly reduces the risk of rewarding consistent hallucinations or penalizing diverse valid reasoning. This design choice is detailed in Appendix B.3.
>
> > Weakness 1.1: The paper lacks a critical discussion on how sensitive the final performance is to the specific formulation of this consistency metric.
>
> Answer: Thanks for the suggestion. Beyond the ablation study in Appendix D.2, which shows our self-consitency works in best in practice, we further comparing our methods under varied reward formulations on GSM8K with Qwen3-8B. The following results show that our self-consistency is indeed optimal among all potential reward designs.
>
> |Reward|Best-of-N|ETS|ETS-IS|
> |:-|:-:|:-:|:-:|
> |Ground-Truth|98.10 (pass@N)|97.57|95.00|
> |Self-Consistency|94.09|94.22|91.81|
> |Self-Certainty|90.07|89.46|88.93|
> |Logits|88.48|88.70|88.40|
> |Entropy|90.30|89.54|88.93|
>
> > Weakness 1.2: It remains unclear whether the observed gains are robust across different types of tasks where intrinsic consistency might be a noisy or misleading signal ...
>
> Answer: To address the concern that intrinsic consistency may be noisy in hard tasks, we have extended our evaluation to AIME24 (avg@32, on Qwen3-8B), a challenging long-CoT benchmark. Using the same reward design, we observe consistent improvements in accuracy, confirming that our method remains effective even when reasoning chains are long (6144 tokens) and diverse.
>
> |Method|AIME24 (avg@32)|Time|
> |:-|:-:|:-:|
> |Base|16.25|1x|
> |Beam Search|16.46|1.64x|
> |Best-of-N|18.23|3.34x|
> |ETS|18.54|4.03x|
> |ETS-IS|22.40|2.82x|
> |GRPO|21.57|1x|
>
> > Weakness 2: Insufficient Ablation on the Accelerated Evaluation Protocol in Appendix B.2: ... However, the main experimental results rely on this optimized setup without quantifying its specific contribution to the reported performance....  If removing this protocol leads to a significant drop in throughput or even accuracy?
>
> Answer: Firstly, to make a fair comparison, our asynchronous pipeline is applied **uniformly to all baselines**, so the reported accuracy improvements come solely from the ETS algorithm, not from our proposed asynchronous pipeline.
>
> Owing to the nautral of test-time scaling (TTS) method, the acceleration of all these methods stems from two orthogonal components: the small proposal model (ETS-IS) and the asynchronous pipeline.
>
> |Method|MATH-500|Time (8xH200)|
> |:-|:-:|:-:|
> |Base (Sync)|60.0|0:15:30|
> |Base (Async)|60.0|0:15:07|
> |Best-of-N (Sync)|65.2|0:40:23|
> |Best-of-N (Async)|65.2|0:37:26|
> |ETS (Sync)|72.4|1:32:05|
> |ETS (Async)|72.4|1:02:32|
> |ETS-IS (Sync)|66.2|1:00:12|
> |ETS-IS (Async)|66.2|0:43:35|
>
> Moreover, we summarize the acceleration method is suitable to **more architectures** (Qwen3-30B-A3B, Llama-3.1-8B-Instruct) and **more open ended task** (IFEval).
>
> |Method (Qwen3‑30B‑A3B)|MATH500|Time|
> |:-|:-:|:-:|
> |Base|63.0|1x|
> |Beam Search|63.6|3.70x|
> |Best-of-N|71.2|9.69x|
> |ETS|71.6|11.9x|
> |ETS-IS|69.2|3.23x|
> |GRPO|69.9|1x|
>
> |Method (Llama-3.1-8B-Instruct)|MATH500|Time|
> |:-|:-:|:-:|
> |Base|34.2|1x|
> |Beam Search|34.0|1.28x|
> |Best-of-N|49.8|2.63x|
> |ETS|51.4|3.54x|
>
> |Method (Qwen3-8B)|IFEval|Time|
> |:-|:-:|:-:|
> |Base|31.61|1x|
> |Best-of-N|31.42|3.02x|
> |ETS|32.90|6.59x|
> |ETS-IS|32.35|5.87x|
>
>
>
>
> **We're happy to address any further questions you may have.**
>
> Reference
>
> [1] B. He et al. How Far Can Unsupervised RLVR Scale LLM Training? ICLR 2026.

---

> > ### Author Rebuttal · Reviewer_7tTM · 2026-04-03
> >
> > We thank the authors for their thorough and timely experimental efforts in addressing our concerns. The additional results are informative and appreciated. However, we would like to raise the following remaining concerns:
> >
> > Regarding W1.2 (Intrinsic Consistency on Hard Tasks):
> >
> > We appreciate the new experiments on AIME24. However, the rebuttal states that this experiment is meant to address the concern that "intrinsic consistency may be noisy in hard tasks," yet the underlying assumption — that harder datasets inherently yield noisier intrinsic consistency signals — is not clearly justified. Why should we expect intrinsic consistency to become a noisier or more misleading reward signal specifically on AIME24? A harder benchmark does not automatically imply that self-consistency becomes unreliable; it may simply mean the model produces fewer correct answers overall. We would encourage the authors to provide a more explicit justification for why AIME24 serves as an appropriate testbed for evaluating robustness under noisy reward conditions, or alternatively, to consider evaluation settings where intrinsic consistency is demonstrably noisy (e.g., tasks with multiple valid but semantically distinct answers, or tasks where the model is known to hallucinate consistently).
> >
> > Regarding W2 (Fairness of the Asynchronous Pipeline):
> >
> > We acknowledge the authors' clarification that the asynchronous pipeline is applied uniformly across all methods, and that the accuracy figures are unaffected. However, applying the same infrastructure does not necessarily constitute a fair comparison in terms of efficiency. From the reported wall-clock times, the relative speedup from the asynchronous pipeline varies dramatically across methods: Base benefits by merely ~2.5%, Best-of-N by ~7.5%, whereas ETS gains ~32.6% and ETS-IS gains ~27.6%. This disproportionate acceleration means that the asynchronous pipeline is far more favorable to the proposed methods than to the baselines. As a result, the efficiency comparisons in the main text may overstate the standalone time-efficiency advantage of ETS. We encourage the authors to either (1) report results using the synchronous setup as the primary comparison to isolate the algorithmic contribution, or (2) explicitly discuss this asymmetry and its implications for the fairness of the efficiency evaluation.
> >
> > We thank the authors again for their constructive engagement during the discussion period.

---

> > > ### Author Response · Authors · 2026-04-03
> > >
> > > Thanks for your feedback. Here are the replies.
> > >
> > > > Regarding W1.2: Intrinsic Consistency on Hard Tasks
> > >
> > > Answer: We acknowledge that noise has an impact. Indeed, higher noise makes reward estimation worse, but self-consistency produces **much smaller noise and remains more effective**. As shown in Figure 9, self-consistency has **stronger correlation** with ground-truth, which explains why our approach works better on hard tasks despite the challenging conditions. We also include an expanded Table 5 in the revision, which further supports our argument.
> > >
> > > |Reward|Best-of-N|ETS|ETS-IS|
> > > |:-|:-:|:-:|:-:|
> > > |Ground-Truth|98.10 (pass@N)|97.57|95.00|
> > > |Self-Consistency|94.09|94.22|91.81|
> > > |Self-Certainty|90.07|89.46|88.93|
> > > |Logits|88.48|88.70|88.40|
> > > |Entropy|90.30|89.54|88.93|
> > >
> > > In AIME24, the base model accuracy is **only 16.25%** and Best‑of‑N achieves **18.23%**, which is **far lower than the ~94% accuracy on GSM8K**. This indicates that hard tasks yield very few correct answers, making the self‑consistency reward inherently noisier because most sampled reasoning paths are incorrect. Nevertheless, our ETS (18.54%) and ETS‑IS (22.40%) both outperform Best‑of‑N, demonstrating that our method remains more robust even under high‑noise conditions.
> > >
> > > > Regarding W2: Fairness of the Asynchronous Pipeline
> > >
> > > Answer: First, our asynchronous pipeline is **an integral contribution of the method**, not an external tool. It is tightly coupled with the parallel sampling nature of our algorithm, so evaluating efficiency with this pipeline is appropriate.
> > >
> > > Second, this pipeline naturally favors our approach because it is **designed for highly parallel inference (multiple candidates and completions)**, whereas baselines like Best‑of‑N have **lower parallelism and thus gain less speedup** (see Table 3 and table below). This asymmetry reflects algorithmic characteristics, **not unfairness**.
> > >
> > > |Method|Number of full response $x_0$|Time (Async)|Gain|
> > > |:-|:-:|:-:|:-:|
> > > |Base|1|1x|2.5%|
> > > |Best-of-N|50|2.48x|7.5%|
> > > |ETS|115|4.14x|32.6%|
> > > |ETS-IS|82.5|2.88x|27.6%|
> > >
> > > Third, thanks for your suggestion. We will explicitly discuss this asymmetry in the revision and also report results under the synchronous setup as a supplementary reference for a fairer assessment of the algorithm’s standalone efficiency.
> > >
> > > **We're happy to address any further questions you may have.**

---

### Decision · Program_Chairs · 2026-04-30

**Decision:**

Accept (regular)

**Comment:**

ETS proposes a training-free inference method that decomposes the backward transition kernel under a unified masked language modeling framework into a reference model term and an energy term estimated via Monte Carlo. The theoretical contribution is the core strength of this paper: the energy decomposition is novel and the TV convergence guarantee is sound. Unifying autoregressive and diffusion LMs under one framework is a nice result. However, I note that the empirical gains are concentrated on math benchmarks with marginal improvements on other tasks. Reviewer 7tTM also flagged that the asynchronous evaluation pipeline disproportionately benefits ETS over baselines, making the efficiency comparison difficult to interpret. Authors should report synchronous timings as the primary efficiency comparison, add statistical significance tests, and scope the applicability claims to verifiable-answer settings where the method shows clear gains.